# Shared features and reciprocal complementation of the *Chlamydomonas* and *Arabidopsis* microbiota

Paloma Durán[1,2,3], José Flores-Uribe[1,3], Kathrin Wippel [1], Pengfan Zhang[1], Rui Guan [1], Barbara Melkonian[1], Michael Melkonian[1] & Ruben Garrido-Oter [1,2✉]

Microscopic algae release organic compounds to the region immediately surrounding their cells, known as the phycosphere, constituting a niche for colonization by heterotrophic bacteria. These bacteria take up algal photoassimilates and provide beneficial functions to their host, in a process that resembles the establishment of microbial communities associated with the roots and rhizospheres of land plants. Here, we characterize the microbiota of the model alga *Chlamydomonas reinhardtii* and reveal extensive taxonomic and functional overlap with the root microbiota of land plants. Using synthetic communities derived from *C. reinhardtii* and *Arabidopsis thaliana*, we show that phycosphere and root bacteria assemble into taxonomically similar communities on either host. We show that provision of diffusible metabolites is not sufficient for phycosphere community establishment, which additionally requires physical proximity to the host. Our data suggest the existence of shared ecological principles driving the assembly of the *A. thaliana* root and *C. reinhardtii* phycosphere microbiota, despite the vast evolutionary distance between these two photosynthetic organisms.

[1] Department of Plant-Microbe Interactions, Max Planck Institute for Plant Breeding Research, 50829 Cologne, Germany. [2] Cluster of Excellence on Plant Sciences, 40225 Düsseldorf, Germany. [3] These authors contributed equally: Paloma Durán, José Flores-Uribe. ✉email: garridoo@mpipz.mpg.de

Plants associate with diverse microbes in their aerial and belowground tissues which are recruited from the surrounding environment. These microbial communities, known as the plant microbiota, provide the host with beneficial functions, such as the alleviation of abiotic stresses[1–4], nutrient mobilization[5–7], or protection against pathogens[8,9]. Characterization of the microbiota associated with a wide range of plant species including liverworts[10], lycopods, ferns[11], gymnosperms[12,13], and angiosperms[14–21] shows a strong influence of host phylogeny as well as conserved and possibly ancestral community features. Furthermore, it has been speculated that the ability to form associations with members of these communities, such as mycorrhizal fungi, was a trait required for the colonization of land by plants 450 Mya, possibly inherited from their algal ancestor[22,23]. In aquatic environments, algae are also known to associate with complex bacterial communities termed phycosphere microbiota[24–27], where exchange of metabolites, including organic carbon[28–31], soluble micronutrients[32], vitamins[33–35], and other molecular currencies[36,37] influence algal growth and development. These features are analogous to those found in the rhizosphere of vascular plants, where secreted diffusible compounds alter soil pH, oxygen availability, the concentration of antimicrobials, and organic carbon, and thus support distinct microbial communities by favoring the growth of certain bacteria while restricting the proliferation of others[25,38–41]. However, it is not known whether the ability to assemble a complex microbiota from the surrounding soil is specific of embryophytes or if extends to other terrestrial green lineages such as chlorophyte and streptophyte algae.

To address this question, we characterized the microbiota of a selection of taxonomically diverse subaerial green algae, including the model organism *Chlamydomonas reinhardtii* (*Cr*), and show significant taxonomic overlap between the root and phycosphere microbiota. To elucidate whether these similarities in community composition also correspond to the functional equivalence between root and phycosphere bacteria, we generated a culture collection of phycosphere bacteria derived from *Cr* and performed cross-inoculation experiments using the model plant *Arabidopsis thaliana* (*At*) and its associated bacterial culture collection[42]. Using synthetic communities (SynComs) we show that the *At* root and *Cr* phycosphere microbiota can be reciprocally complemented with bacteria derived from the other host. Finally, we show that physical proximity between *Cr* and its microbiota is required for the establishment of phycosphere communities, suggesting that this process is not merely a consequence of the provision of diffusible photosynthates by the host.

## Results

### *Cr* assembles a distinct microbiota from the surrounding soil.
To determine whether *Cr* shapes soil-derived bacterial communities similarly to land plants, we designed an experiment where *At* and *Cr* were grown in parallel in natural soil in the greenhouse (experiment A; Supplementary Fig. 1a and Supplementary Table 1). Briefly, pots containing Cologne Agricultural Soil (CAS) were inoculated with axenic *Cr* (CC1690) cultures or sowed with surface-sterilized *At* (Col-0) seeds. We then collected samples from unplanted controls and from the surface of *Cr*-inoculated pots (phycosphere fraction) at 7-day intervals, and harvested the root and rhizosphere of *At* plants after 36 days (Methods). Bacterial communities from all compartments were characterized by 16S rRNA amplicon sequencing. Analysis of community profiles showed a decrease in α-diversity in the phycosphere and root compartments compared to the more complex soil and rhizosphere communities (Fig. 1a). In addition, analysis of β-diversity revealed a significant separation by compartment, where phycosphere and root samples formed

distinct clusters that were also separated from those consisting of soil and rhizosphere samples (Fig. 1b; 22.4% of variance; $P < 0.001$). Further inspection of amplicon profiles showed overlap between root- and phycosphere-associated communities along the second and third components (Fig. 1c), suggesting similarities between the bacterial communities that associate with *Cr* phycospheres and *At* roots. Furthermore, an analysis where we simulated the process of recruitment of bacterial species from the starting soil microbiota by *Cr* using the hypergeometric distribution shows that the observed overlap between the *Cr* phycosphere and *At* root communities (Fig. 1c, d) is highly significant ($P = 3.06 \times 10^{-33}$) and cannot be explained by a random process of microbial recruitment from the same starting soil community.

To characterize the dynamics of these microbiota assembly processes, we analyzed the time-series data from soil and phycosphere and end-point community profiles from *At* roots. This revealed gradual recruitment of bacterial taxa from the soil, leading to the formation of distinct phycosphere communities that become significantly differentiated 21 days after inoculation, which is comparable to that observed in *At* root-associated communities at day 36 (Supplementary Fig. 2a). Subsequent enrichment analysis of amplicon sequence variants (ASVs) in each compartment, compared to unplanted soil, showed an increase in the relative abundance of *Cr*- and *At*-enriched ASVs in phycosphere and root samples, respectively. In contrast, the total relative abundance of soil-enriched ASVs progressively decreased in host-associated compartments, while remaining stable in unplanted soil (Supplementary Fig. 2b–d). Although the magnitude of the changes in bacterial community composition in the phycosphere diminishes over time, it remains unclear whether these communities reach a steady state over the duration of the experiment. Taken together, these results indicate that, similarly to *At*, *Cr* is able to recruit a subset of bacterial taxa from the surrounding soil and assemble a distinct microbiota.

To test whether the assembly of phycosphere communities from soil could also be observed as a result of the proliferation of naturally occurring subaerial microalgae, we characterized the eukaryotic and prokaryotic microbial communities from natural soil. Briefly, CAS soil was irrigated and placed in sterile microboxes in growth chambers with a day/dark-light cycle (L/D 12/12) and incubated for 7 weeks. At the end of the experiment, we obtained samples from the soil surface, measured chlorophyll content, and performed eukaryotic 18S and bacterial 16S rRNA profiling. As negative controls, we obtained samples from the initial soil, as well as from soil that was kept in the dark but otherwise under the same conditions throughout the duration of the experiment (experiment B; Supplementary Fig. 1b and Supplementary Table 1). We observed a significant increase of chlorophyll fluorescence in the soil surface exposed to light compared to the negative controls, which was concomitant with a significant increase in the relative abundance of chlorophyte algae in the soil surface when exposed to light (Supplementary Fig. 3a, b). While fungi were the most abundant eukaryotes in the initial soil samples, chlorophytes dominated the soil surface communities at the end of the experiment and were significantly more abundant in the light-treated samples compared to the dark controls. Interestingly, we observed that *Chlamydomonas* was one of the most abundant eukaryotic genera in the light-exposed soil, and included several ASVs assigned to *Cr*. Together with this significant proliferation of green algae, we observed differentiation of the bacterial communities of light-exposed soil surface samples compared to those from the initial soil and negative controls, which clustered together (Supplementary Fig. 3d). A comparison with 16S profiles obtained from the greenhouse experiment (Fig. 1) revealed that the communities found in the

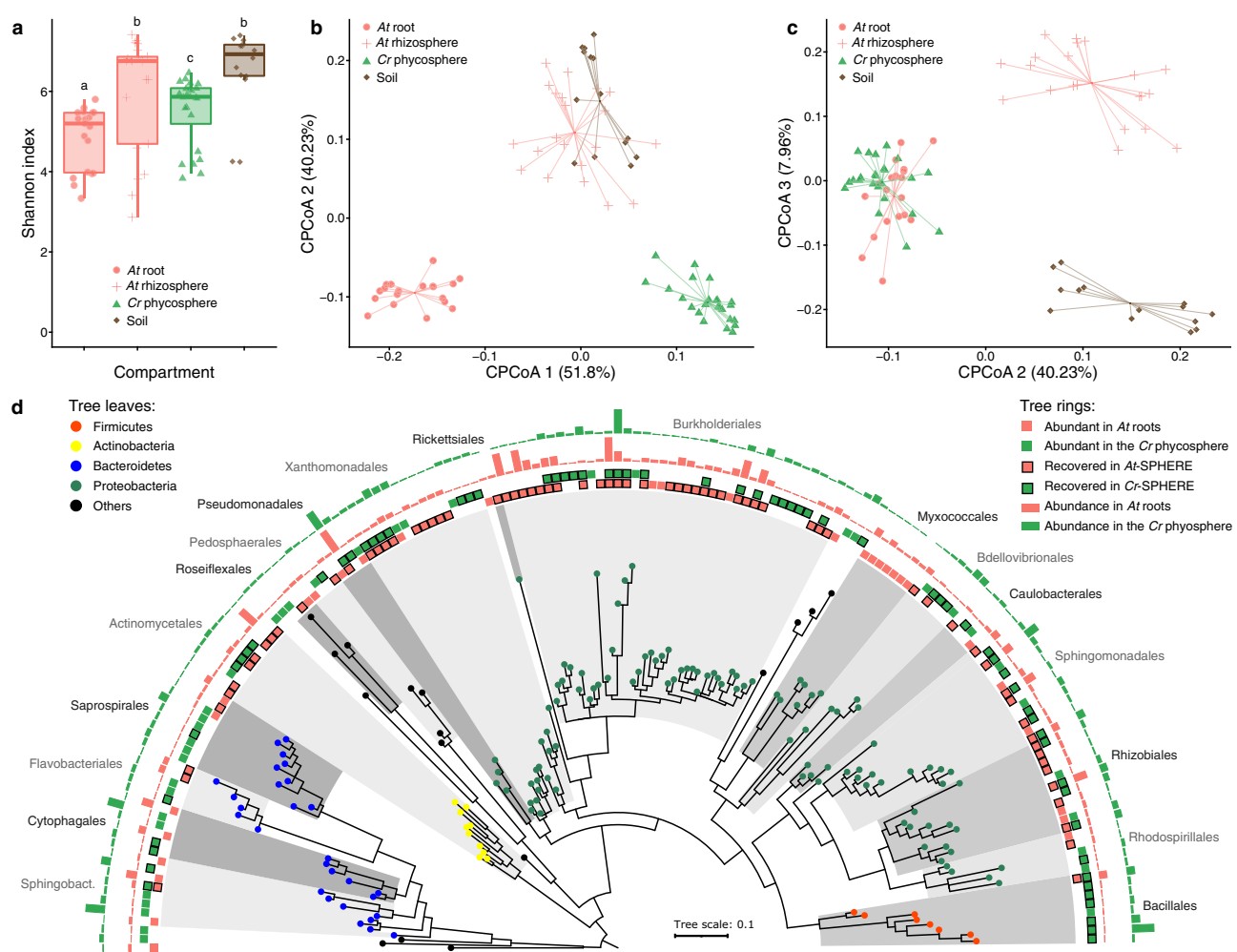

**Fig. 1 Comparison of bacterial community structures associated with *At* roots and the *Cr* phycosphere in natural soil. a** Alpha diversity estimates of bacterial communities from soil ($n = 14$), rhizosphere ($n = 21$), root ($n = 18$), and phycosphere ($n = 24$) samples from *At* and *Cr* grown in CAS soil in the greenhouse. A Kruskal–Wallis test followed by a Dunn's post hoc was used to assess significant differences among groups ($P < 0.05$). **b, c** PCoA of Bray–Curtis dissimilarities (ASV-level) constrained by compartment (22.4% of variance explained; $P < 0.001$). **d** Phylogeny of partial 16S rRNA sequences of the most abundant bacterial OTUs found in *At* roots ($n = 120$), and *Cr* phycosphere ($n = 141$) community profiles. Leaf nodes are colored by taxonomic affiliation (phylum level). The two innermost rings (colored squares) represent abundant OTUs in each compartment. Squares highlighted with a black contour correspond to OTUs for which at least one representative bacterial strain exists in the IRL or IPL culture collections. The two outermost rings (barplots) represent log-transformed relative abundances of each OTU in *At* root or *Cr* phycosphere samples. Corresponds to experiment A (Supplementary Fig. 1a and Supplementary Table 1).

light-exposed soil surface resembled those from the *Cr* phycosphere and, together with samples from the rhizosphere and root of *At*, separated from the initial soil and dark soil surface samples along the second principal component (Supplementary Fig. 3c).

**The *Cr* phycosphere and the plant root share a core microbiota.** Given the observed similarities between phycosphere and root communities (Fig. 1c), we compared the most abundant taxonomic groups found in association with the two photosynthetic hosts. We found a significant overlap between Operational Taxonomic Units (OTUs) with the highest relative abundances in either phycosphere or root samples (Fig. 1d; >0.1% relative abundance; 32% shared; $P < 0.001$), which included members of every bacterial order except Myxococcales, which were only found in large relative abundances in *At* root samples (Supplementary Data 1). In line with previous descriptions of the *At* root microbiota, we observed that these host-associated communities were dominated by Proteobacteria, and also included members of the Actinobacteria, Bacteroidetes, and Firmicutes phyla. At this

taxonomic level, the major difference between the two photosynthetic hosts was given by a lower contribution of Actinobacteria and a larger relative abundance of Firmicutes in the *Cr* phycosphere compared to the *At* root compartment (Fig. 1d). Given that this latter phylum is most abundant in soil, this difference may be due to the difficulty of fully separating soil particles from the phycosphere fraction during sample collection.

Next, we sought to assess whether the observed overlap in community structures between *Cr* and *At* could be extended to other land plant lineages. We performed a meta-analysis, broadening our study to include samples from phylogenetically diverse plant species found in a natural site, including lycopods, ferns, gymnosperms, and angiosperms[11], as well as the model legume *Lotus japonicus* (*Lj*) grown in CAS soil in the greenhouse[7,43]. First, we determined which taxonomic groups were present in each plant species (≥80% occupancy and ≥0.1% average relative abundance) and identified a total of six bacterial orders that consistently colonize plant roots (i.e., found in every host species). These taxa include Caulobacterales, Rhizobiales, Sphingomonadales, Burkholderiales, Xanthomonadales (Proteobacteria), and

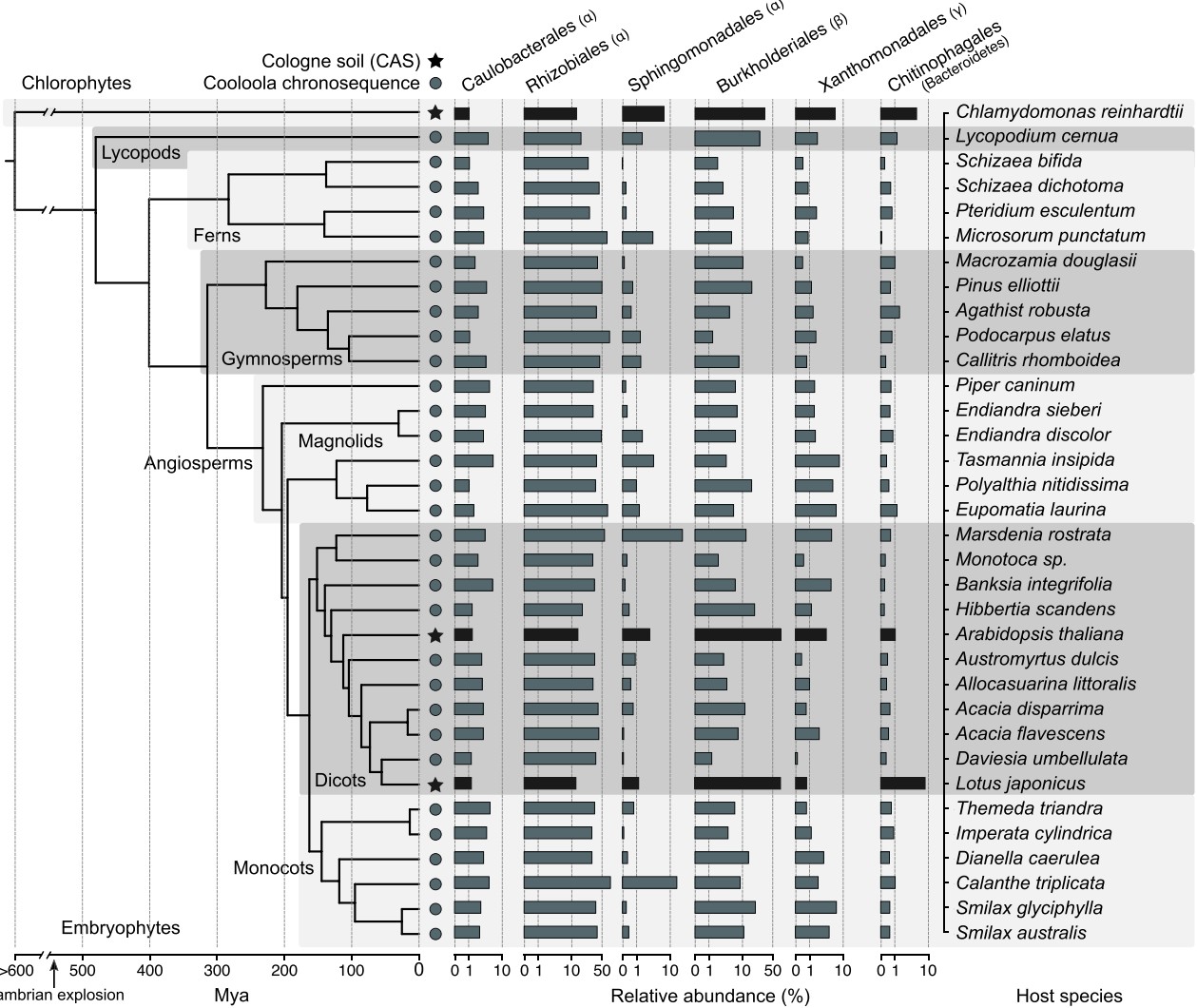

**Fig. 2 Conservation of bacterial orders of the root and phycosphere microbiota across photosynthetic organisms.** Phylogeny was inferred from a multiple sequence alignment of the ribulose-bisphosphate carboxylase gene (*rbcL*) of 35 plant species and *Chlamydomonas reinhardtii*. The bar plots represent the average aggregated relative abundance of the six bacterial orders found to be present in the root microbiota of each plant species (80% occupancy and ≥0.1% average relative abundance). Leaf nodes depicted with a star symbol denote community profiles of plants grown in CAS soil in the greenhouse[7,43], whereas those marked with a circle were obtained from plants sampled at the Cooloola natural site chronosequence[11]. Includes data for *At* and *Cr* from experiment A (Supplementary Fig. 1a and Supplementary Table 1).

Chitinophagales (Bacteroidetes). We observed that the aggregated relative abundance of these six bacterial orders accounted for 39% of their respective communities on average (Fig. 2). Interestingly, these taxa were also found among the most abundant in the phycosphere of *Cr* (45% aggregated relative abundance), as well as in association with the native CAS soil algae (33%; Supplementary Fig. 3c). These results indicate that members of the core microbiota of land plants also associate with subaerial chlorophyte algae and form part of phycosphere communities in terrestrial environments.

**Reconstitution of phycosphere communities using reductionist approaches**. After the characterization of phycosphere-associated bacterial communities in natural soil, we sought to develop systems of reduced complexity that would allow controlled perturbation of environmental parameters, and targeted manipulation of microbial community composition. First, we established a mesocosm system using soil-derived microbial communities as start inocula (experiment C; Supplementary Fig. 1c and Supplementary Table 1). We co-inoculated axenic *Cr* (CC1690) cultures

with microbial extracts from two soil types (CAS and Golm) in two different carbon-free media (TP and B&D), which ensures that the only source of organic carbon to sustain bacterial growth is derived from *Cr* photosynthetic activity. These phycosphere mesocosms were then incubated under continuous light for 11 days, during which we assessed *Cr* growth using cell counts, and profiled bacterial communities via 16S rRNA amplicon sequencing. In this system, *Cr* was able to steadily grow without a detrimental impact from co-inoculation with soil-derived bacterial extracts (Supplementary Fig. 4a). Analysis of diversity showed that *Cr* was able to shape soil-derived bacterial communities within the first 4 days, compared to the starting inocula, and that these phycosphere communities remained stable until the end of the experiment (Fig. 3). Interestingly, cultivation of soil-derived bacteria in the absence of organic carbon or supplemented with artificial photosynthates (AP) led to significantly differentiated bacterial communities (Fig. 3a; 17.9% of the variance; $P < 0.001$). In addition, inoculation of soil-derived bacteria with heat-killed *Cr* cultures was not sufficient to recapitulate this community shift (Supplementary Fig. 4b), suggesting that the

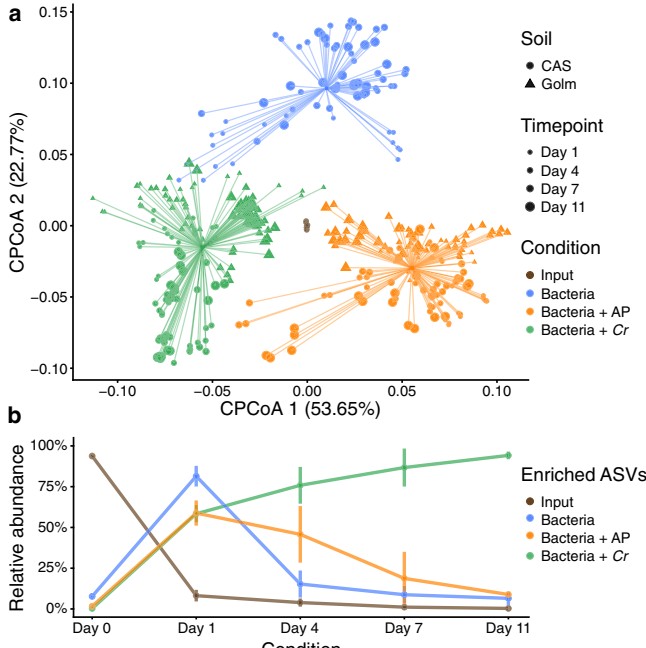

**Fig. 3 Mesocosm experiments recapitulate the establishment of phycosphere communities by *Cr* across soil types and growth media.** **a** PCoA analysis of Bray–Curtis dissimilarities of bacterial community profiles (ASV-level), constrained by condition (17.9% of the variance; *P* < 0.001), showing a significant separation between start inocula (soil washes, depicted in brown; *n* = 12), phycosphere communities (green; *n* = 158), and soil washes incubated in minimal media (blue; *n* = 112), or media supplemented with artificial photoassimilates (APs, depicted in orange; *n* = 144). **b** Dynamic changes in the phycosphere community composition in terms of the aggregated relative abundances of ASVs enriched in each condition (*n* = 158) with respect to the start inocula (*n* = 12). Error bars represent the standard deviation of the data. Corresponds to experiment C (Supplementary Fig. 1c and Supplementary Table 1).

presence of live and metabolically active *Cr* is required for the establishment of synthetic phycospheres. A separate experiment where microbial inocula were filtered through a 5 μm pore-size membrane showed similar bacterial community shifts compared to non-filtered extracts (Supplementary Fig. 4c), suggesting that the presence of larger eukaryotic microorganisms in the soil microbial extracts did not contribute to the observed changes in bacterial composition. Similar to the results obtained using natural soil, the aggregated relative abundance of *Cr*-associated ASVs in the synthetic phycosphere samples increased over time, whereas ASVs enriched in the bacteria-only control samples consistently decreased (Fig. 3b). At the end of the experiment (day 11), the relative abundance of *Cr*-enriched ASVs accounted for 94% of the entire phycosphere community, in contrast to a lower contribution observed in the natural soil system (Supplementary Fig. 2b; 60% relative abundance at day 36). This pattern could be a consequence of the unintended depletion of bacteria that are not capable of metabolizing *Cr*-secreted photoassimilates in a liquid environment, and in these specific culture media. Finally, an independent mesocosm experiment using day/night light cycles showed delayed but similar patterns to those using continuous light, indicating that phycosphere community establishment may be independent of *Cr* culture synchronization (Supplementary Fig. 4d).

To test whether the process of phycosphere microbiota assembly is also found in other taxonomic lineages of green algae besides the chlorophyte *Cr*, we performed a similar mesocosm experiment (experiment D; Supplementary Fig. 1d and Supplementary Table 1) where we characterized the soil-derived phycosphere microbiota of a selection of subaerial algal strains isolated from the same terrestrial environment. This group of algae includes chlorophytes from the classes Trebouxiophyceae (*Microthamnion*), and Chlorophyceae (*Chlamydomonas*), as well as streptophytes belonging to the Klebsormidiophyceae (*Klebsormidium*), and Zygnematophyceae (*Spiroglea*) classes, the latter of which belongs to the genus that likely represents the closest extant relative to the most recent common ancestor of streptophyte algae and embryophytes[44]. Analysis of 16S rRNA amplicon profiles shows that these microalgae can assemble distinct soil-derived microbial communities, in a pattern of diversification that relates to the host phylogeny (Supplementary Fig. 5a, b). Interestingly, the phycosphere community of the *Cr* strain CC1690 was closest to that of the *Chlamydomonas* sp. environmental isolate (MEL 1030 B), indicating that this laboratory strain interacts with soil-derived communities in a similar manner as an environmental strain adapted to living within these complex communities. Furthermore, the phycospheres of all characterized algal strains were dominated by members of the core bacterial taxa shared with the root microbiota of land plants (Supplementary Fig. 5c). The results from this experiment indicate that the assembly of distinct phycosphere communities from the soil is not specific for *Cr* but can be also observed for other subaerial green algae from diverse taxonomic groups.

Next, we aimed to control community composition in this reductionist system by establishing a *Cr*-associated bacterial culture collection following a similar approach as reported in previous studies with land plants[42,45–49]. We employed a limiting dilution approach using 7 day-old *Cr* phycospheres derived from CAS soil bacteria incubated in two minimal media (TP and B&D; Supplementary Fig. 1e). The resulting sequence-indexed phycosphere bacterial library (*Cr*-IPL) contained a total of 1645 colony-forming units, which were taxonomically characterized by 16S rRNA amplicon sequencing. Comparison of these sequencing data with the community profiling of soil phycospheres revealed that we were able to recover 62% of the most abundant bacterial OTUs found in natural communities (Supplementary Fig. 6a and Data 2). Recovered OTUs accounted for up to 63% of the cumulative relative abundance of the entire culture-independent community, indicating that our collection is taxonomically representative of *Cr* phycosphere microbiota. These results are comparable to the recovery rates observed in previously reported culture collections from different plant species (e.g., 57% for *A. thaliana*[42]; 69% for rice[6]; 53% for *L. japonicus*[48]).

To establish a core collection of phycosphere bacteria, we selected a taxonomically representative set of strains from the *Cr*-IPL covering all major taxonomic groups found in the culture-independent community profiles and subjected them to whole-genome sequencing. In total, we sequenced the genomes of 185 bacterial isolates, classified into 42 phylogroups (97% average nucleotide identity), belonging to 5 phyla and 15 families (Supplementary Data 3). Next, we performed comparative analyses of the genomes from the phycosphere core collection (*Cr*-SPHERE) with those established from the soil, roots of *A. thaliana*, and roots and nodules of *L. japonicus* (At- and Lj-SPHERE) grown in the same soil (CAS). The whole-genome phylogeny of these bacterial strains showed that all major taxonomic groups that included root-derived isolates were also represented in the *Cr*-SPHERE collection, but not in the soil collection (Supplementary Fig. 6b). Importantly, the phycosphere

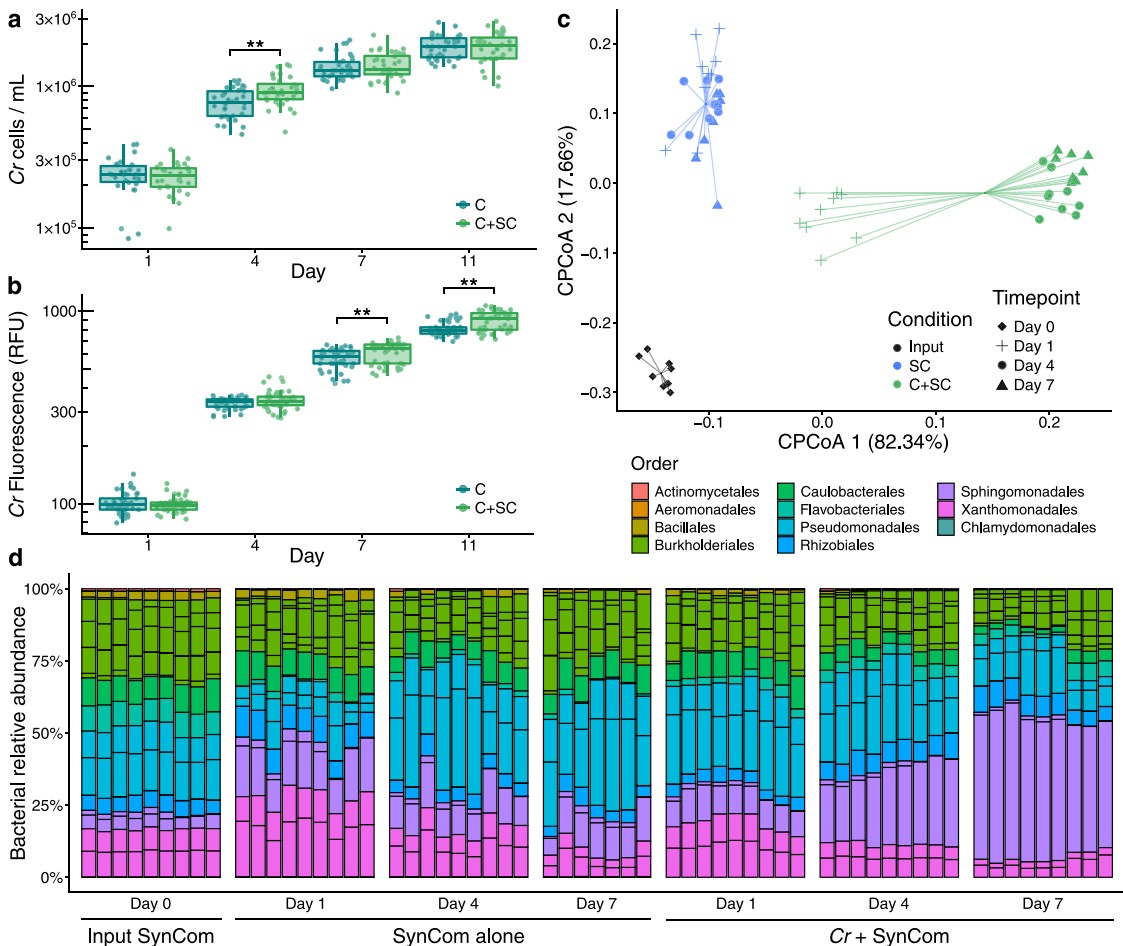

**Fig. 4 Phycosphere reconstitution using bacterial SynComs derived from the *Cr*-SPHERE core culture collection.** *Cr* growth in the gnotobiotic system axenically (*n* = 36) or in co-inoculation with the bacterial SynCom (*n* = 36), measured as algal cell densities (**a**), and chlorophyll fluorescence (**b**). A Mann–Whitney test was used to assess significant differences among groups (FDR-corrected; *P* < 0.05). **c** Strain-level beta-diversity analysis (CPCoA of Bray–Curtis dissimilarities; 40.4% of the variance; *P* < 0.001) of bacterial communities from samples obtained from a liquid-based gnotobiotic system. Samples are color-coded based on the experimental condition: input SynCom samples (black; *n* = 9), synthetic phycospheres (light green; *n* = 27), and SynCom only controls (blue; *n* = 25). **d** Bar charts showing relative abundances of individual SynCom members across conditions and time points. Corresponds to experiment F (Supplementary Fig. 1f and Supplementary Table 1).

collection also included multiple representatives of each of the six bacterial orders that were found to consistently colonize plant roots in natural environments (Fig. 2). Next, we assessed the functional potential encoded in the genomes of the sequenced phycosphere bacteria using the KEGG orthology database as a reference[50]. Principal coordinates analysis of functional distances showed that bacterial taxonomy accounted for most of the variance of the data (58.63%; *P* < 0.001), compared to a much smaller impact of the host of origin of the genomes (4.22% of the variance; *P* < 0.001; Supplementary Fig. 6c).

Next, we tested whether synthetic communities formed by isolates from the *Cr*-SPHERE collection could recapitulate assembly patterns of natural phycospheres under laboratory conditions. Axenic *Cr* cultures (CC1690) were inoculated with a bacterial SynCom composed of 26 strains that could be distinguished at the 16S level and contained representative members of all major phycosphere taxonomic groups (experiment F; Supplementary Fig. 1f, Table 1 and Data 4). Assessment of *Cr* growth using chlorophyll fluorescence and cell counts showed that the presence of the bacterial SynCom had no consistent beneficial or detrimental impact on *Cr* proliferation in this system (Fig. 4a, b), similarly to what we observed in mesocosms (Supplementary Fig. 4a). Analysis of time-course

amplicon profiles showed that *Cr* assembled a characteristic phycosphere community within the first 4 days of co-inoculation which was significantly separated from both start inocula and bacterial SynComs alone (Fig. 4c, d). Together, these results demonstrate that we can recapitulate the *Cr* assembly of distinct phycosphere communities in natural soils using culture-dependent and -independent gnotobiotic systems.

**_Cr_- and *At*-derived SynComs form taxonomically similar communities on either host.** Given the similarity between phycosphere and root communities observed in natural soils (Fig. 1), and the taxonomic and functional overlap across genomes from their corresponding core collections (Supplementary Fig. 6b, c), we hypothesized that SynComs with the same taxonomic composition would assemble into similar communities, regardless of their origin. To test this hypothesis, we used a soil-based gnotobiotic system in which we could grow *Cr* and *At* in parallel, in addition to the previously described liquid-based system. We designed taxonomically-paired SynComs composed of strains from either the IPL (*Cr*-SPHERE) or IRL (*At*-SPHERE) bacterial culture collections. In these SynComs we included one representative strain from each bacterial family shared between the two

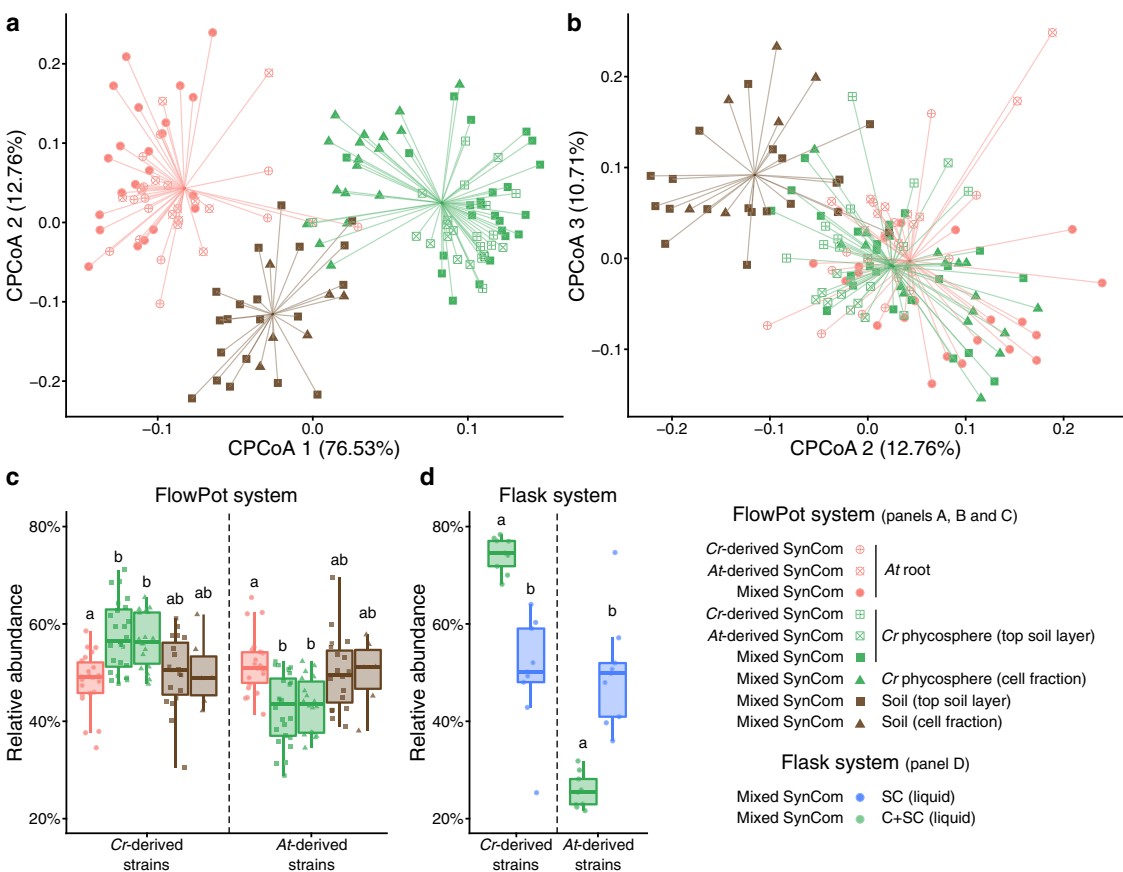

**Fig. 5 Root and phycosphere bacteria colonize *At* and *Cr* and assemble into taxonomically equivalent communities. a** Strain-level beta diversity analysis of soil ($n = 26$), root ($n = 57$), and phycosphere ($n = 66$) bacterial community profiles, from gnotobiotic *At* and *Cr*, inoculated with bacterial SynComs derived from *At* roots (*At*-SPHERE), *Cr* (*Cr*-SPHERE), or mixed (*At*- and *Cr*-SPHERE), grown in the FlowPot system. **b** CPCoA of Bray–Curtis dissimilarities of the same samples, aggregated at the bacterial family level (6.4% of the variance; $P < 0.001$). **c, d** Aggregated relative abundances of *At*- and *Cr*-derived strains in the mixed SynCom in a soil-derived ($n = 149$, **c**), and liquid-based ($n = 18$, **d**) gnotobiotic system. A Kruskal–Wallis test followed by a Dunn's post hoc was used to assess significant differences among groups ($P < 0.05$). Corresponds to experiment G (Supplementary Fig. 1g and Supplementary Table 1).

collections ($n = 9$), ensuring that they could be differentiated by their 16S rRNA sequences (Supplementary Data 4). We then inoculated axenic *Cr* cultures and *At* seeds with either IPL, IRL, or mixed (IPL + IRL) SynComs and allowed them to colonize either host for five weeks (experiment G; Supplementary Fig. 1g and Supplementary Table 1). Next, we harvested the root, soil, and phycosphere fractions, measured host growth, and performed 16S rRNA amplicon sequencing. Assessment of growth parameters (cell counts for bacteria and *Cr*, chlorophyll content for *Cr*, and shoot fresh weight for *At*) showed no significant differences across SynCom treatments (Supplementary Fig. 7). However, analysis of community profiles of the mixed SynComs showed that *Cr* and *At* assemble distinct communities that could also be clearly separated from unplanted soil (Fig. 5a). Similar to what we observed in natural soil (Fig. 1c), there was an overlap between phycosphere and root samples, which clustered together along the second and third components (Fig. 5b). Interestingly, analysis of community composition at the family level showed that all SynComs (*Cr*-, *At*-derived, and mixed) formed taxonomically indistinguishable root or phycosphere communities, independently of their host of origin (Fig. 5b). These results indicate that the root and rhizosphere microbiota of axenic *At* or *Cr*, respectively, can be reciprocally complemented with bacterial SynComs derived from either host. Furthermore, analysis of aggregated relative abundances from mixed communities showed that phycosphere-derived strains could successfully colonize *At* roots (48.32% relative abundance), and root-derived strains

established associations with *Cr* in both soil and liquid systems (42.94% and 25.70% relative abundance, respectively; Fig. 5c, d). Despite this capacity for ectopic colonization, we observed significant signatures of host preference in SynComs from the two culture collections, indicated by the fact that *Cr*-derived strains reached higher aggregated relative abundances in the phycosphere compared to the root, while the opposite pattern was identified for *At*-derived bacteria (Fig. 5c). This tendency was accentuated in the liquid system, where *Cr* bacteria outcompeted *At* strains in the presence of the algae but not when they have incubated alone (Fig. 5d). Taken together, these results suggest the presence of conserved features in bacterial members of the *Cr* and *At* microbiota at a high taxonomic level, with signatures of host preference at the strain level.

**Physical proximity is required for the assembly of phycosphere communities and promotion of *Cr* growth.** Next, we sought to investigate whether the observed formation of distinct phycosphere communities is driven by the secretion of diffusible photoassimilates and to what extent physical proximity to bacteria is required to establish other forms of interactions. To test this hypothesis, we developed a gnotobiotic split co-cultivation system where synthetic phycospheres could be grown photoautotrophically (experiment H; Supplementary Fig. 1h and Supplementary Table 1). In this system, two growth chambers were connected through a 0.22 μm-pore polyvinylidene fluoride (PVDF) membrane that allows diffusion of compounds but not

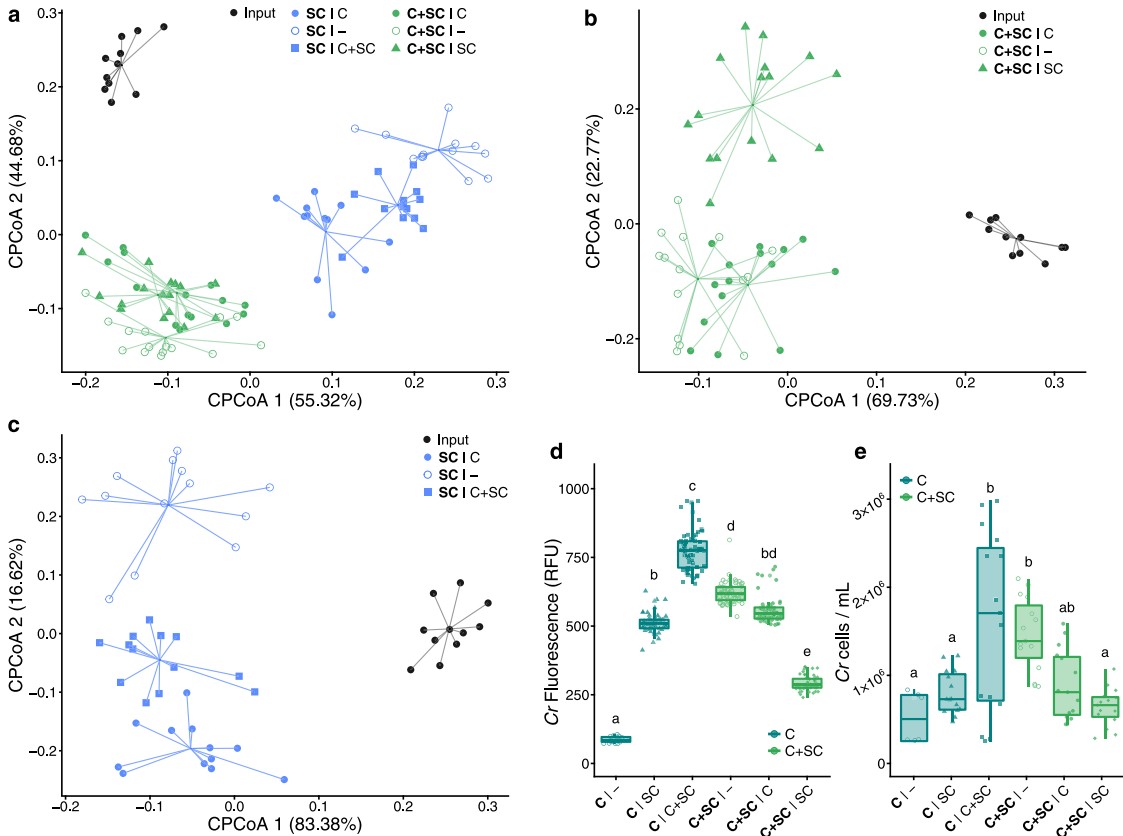

**Fig. 6 Physical proximity to *Cr* is required for the establishment of phycosphere bacterial communities.** Strain-level beta-diversity analyses of Bray–Curtis dissimilarities of bacterial SynComs (SC, *n* = 37), and synthetic phycospheres (SC + C, *n* = 46), grown in a split gnotobiotic system. Constrained PCoA is shown for all samples (21% of variance; *P* < 0.001, **a**), or a subset of samples, depending on the content of the vessel (39.4–39.5% of variance; *P* < 0.001, **b**, **c**). **d**, **e** *Cr* growth across conditions measured using relative chlorophyll fluorescence (*n* = 60, **d**) and algal cell densities (*n* = 15, **e**). A Kruskal–Wallis test followed by a Dunn's post hoc was used to assess significant differences among groups (*P* < 0.05). Corresponds to experiment H (Supplementary Fig. 1h and Supplementary Table 1).

the passage of bacterial or algal cells (Methods). We co-cultivated axenic *Cr* cultures (C), bacterial SynComs (SC), and synthetic phycospheres (C + SC) in these split chambers containing minimal carbon-free media (TP) in multiple pair-wise combinations (Supplementary Data 4). Analysis of 16S rRNA amplicon profiles after 7 days of incubation revealed that SC and C + SC samples were distinguishable from the input bacterial SynComs (Fig. 6a). In addition, samples clustered according to the presence of *Cr* in the same compartment, causing SC and C + SC samples to be significantly separated, independently of the community present in the neighboring chamber (Fig. 6a, indicated by colors; 21.4% of the variance; *P* < 0.001). We observed that differences between the **C + SC**|– and **SC**|C communities in the split system were mainly explained by a significant enrichment of Sphingomonadaceae and Comamonadaceae when the SynCom was growing in the same compartment as *Cr* (C + **SC**|–), and depletion of Pseudomonadaceae. Despite differences among experimental systems, these patterns mirror the results from the natural soil, mesocosm, and SynCom flask experiments (Figs. 1–4), where Sphingomonadaceae was one of the most abundant taxa in the *Cr* phycosphere and Pseudomonadaceae dominated the bacterial communities in the absence of *Cr*.

Comparison of amplicon profiles of samples taken from chambers containing C + SC further showed a significant impact of the content of the neighboring compartment on community structures (Fig. 6b, indicated by shapes; 39.5% of the variance; *P* < 0.001). Interestingly, we also observed that the presence of *Cr* in the neighboring compartment was sufficient to change SC

communities where the bacterial SynCom was incubated alone (Fig. 6c; **SC**|C or **SC**|C + SC vs. **SC**|–; *P* = 0.001), possibly by secreting diffusible compounds or inducing changes in the composition of the culture medium (e.g., minerals and pH). Furthermore, SC communities where *Cr* was present in the neighboring compartment could be differentiated depending on whether *Cr* was in direct contact with bacteria or grown axenically (Fig. 6c; **SC**|C vs. **SC**|C + SC). These community shifts could be explained by competition for diffusible metabolites with the neighboring compartment containing the SynCom together with the algae (C + SC), or by physiological changes in *Cr* induced by physical proximity with bacteria.

In parallel to bacterial community profiles, we assessed *Cr* growth by measuring chlorophyll fluorescence and algal cell counts in all vessels. We observed significant differences in the growth of axenic *Cr* cultures depending on the contents of the neighboring chamber, where the bacterial SynCom alone (C|SC) had a positive impact on the microalgae compared to the control (C|–; Fig. 6d, e). Remarkably, the presence of a synthetic phycosphere in the neighboring compartment had the strongest positive impact on axenic *Cr* cultures (C|C + SC; Fig. 6d, e), suggesting that changes in bacterial community composition driven by physical proximity to *Cr* lead to a beneficial impact on algal growth. In addition, chlorophyll fluorescence and cell counts of synthetic phycospheres (C + SC) were higher when no other microorganisms were incubated in the neighboring chamber (C + SC|– *versus* C + SC|C or C + SC|SC; Fig. 6d, e), possibly due to competition for diffusible compounds. An additional full-

factorial replicate experiment using a modified version of this split co-cultivation system showed consistent results both in community structures and *Cr* growth parameters (Supplementary Fig. 8), despite a large technical variation in cell density measurements (Fig. 6d). Together, these results indicate that the physical proximity of bacteria to *Cr* is required for assembly and growth of phycosphere communities, which in turn may benefit host growth by providing metabolites and/or other compounds including carbon dioxide, which in this experimental setup is likely limiting autotrophic growth of *Cr*. Future experimentation with synthetic phycospheres composed by SynComs designed using combinatorial approaches, coupled with metabolomic and transcriptomic profiling, will be needed to decipher the molecular and genetic mechanisms driving these interactions, and how they compare to the interactions between land plants and their microbiota.

## Discussion

Microscopic algae release photoassimilated carbon to the diffusible layer immediately surrounding their cells, which constitutes a niche for heterotrophic bacteria. Microbes from the surrounding environment compete for colonization of this niche and assemble into complex communities that play important roles in global carbon and nutrient fluxes. These ecological interactions have been well studied in aquatic environments, where each year approximately 20 Gt of organic carbon fixed by phytoplankton are consumed by heterotrophic bacteria[28], which can account for up to 82% of all algal-derived organic matter[51]. For multiple species of green algae, optimal growth, in turn, requires interactions with their associated phycosphere bacteria, which can provide beneficial services to their hosts, such as mobilization of non-soluble iron[32], or exogenous biosynthesis of organic compounds such as vitamins[33,35]. Despite the known importance of these interactions in marine environments, the role of algae-bacterial associations in terrestrial ecosystems remains understudied. This gap in our understanding could be explained by the fact that aquatic phytoplankton is more readily noticed and more amenable to systematic study compared to edaphic microalgae. However, exploring the role of soil-borne unicellular photosynthetic organisms as hosts of complex microbial communities could expand our understanding of carbon and energy fluxes in terrestrial ecosystems.

The results from our culture-independent and gnotobiotic experiments using the model alga *Cr*, which was originally isolated from soil[52], illustrate that this chlorophyte alga can recruit and sustain the growth of heterotrophic, soil-borne bacteria. By characterizing interactions with soil bacterial communities using a panel of subaerial algae isolated from a natural site (Supplementary Fig. 5), we show that this process of recruitment can be also found in other chlorophyte and streptophyte algae. Our in-depth characterization of the *Cr* microbiota reveals clear differences as well as striking similarities in the taxonomic affiliation of abundant root and phycosphere community members (Fig. 1d). Notably, these similarities are found despite fundamental differences between *At* and *Cr*, such as in cell wall composition, (e.g., complex polysaccharides such as cellulose in plant roots, or (glycol)proteins in *Cr*[53]), or the fact that *Cr* secretes a range of fermentation products at night[54]. Among the bacterial lineages shared between the root and phycosphere microbiota, we found groups that are known to establish intimate interactions with multicellular plants, ranging from symbiotic to pathogenic, such as Rhizobia, Pseudomonas, Burkholderia, or Xanthomonas[47,55–57]. Meta-analyses of available data from multiple studies further confirm this pattern by revealing the presence of a set of six bacterial orders, found as abundant members not only in the root communities of all analyzed embryophytes, but also in the phycospheres of several chlorophyte

and streptophyte algae, including *Cr* (Supplementary Figs. 3 and 5), and in associations with naturally occurring microalgal communities (Supplementary Fig. 3). These findings suggest that the capacity to associate with a wide range of photosynthetic organisms is a common trait of these core bacterial taxa, which might predate the emergence of more specialized forms of interaction with their host. This hypothesis was implicitly tested in our cross-inoculation gnotobiotic experiments, where bacterial strains originally isolated from the roots of *At* or the phycosphere of *Cr* competed for colonization of either host (Supplementary Fig. 1g). The observation that *Cr*-derived strains could colonize *At* roots in a competition setup, and *At*-derived bacterial SynComs also populated *Cr* phycospheres (Fig. 5c) supports the existence of shared bacterial traits for establishing general associations with photosynthetic hosts. Despite these patterns of ectopic colonization, we also detected significant signatures of host preference, illustrated by the observation that native bacterial SynComs outcompeted non-native strains in the presence of either host, but not in their absence (Fig. 5c). These findings are in line with a recent comparative microbiota study where similar results were observed for bacterial commensals from two species of land plants (*A. thaliana* and *L. japonicus*[48]). In addition, SynComs composed of strains exclusively derived from the *At*- or the *Cr*-SPHERE collections assembled into taxonomically similar communities on either host, which were indistinguishable at the family level (Fig. 5b). Thus, we show the microbiota of axenic *At* or *Cr* can be reciprocally complemented with bacterial SynComs derived from either host. These data indicate that elements of the mechanism by which the microbiota of photosynthetic organisms are assembled are likely shared between the chlorophyte alga *Cr*, and the embryophyte *At*, despite their vast evolutionary distance.

Organic carbon is assumed to be the main factor limiting bacterial growth in soil[58]. Thus, secretion of organic carbon compounds by photosynthetic organisms constitutes a strong cue for the assembly of soil-derived microbial communities[39,59,60]. The observed similarities between the *At* root and *Cr* phycosphere microbiota at a high taxonomic level, and the fact that, unlike flowering plants, unicellular algae most likely lack a complex immune system, suggest that the release of photoassimilates acts as a first organizing principle driving the formation of these communities. This hypothesis is also supported by a recent study with marine bacterial mesocosms where community composition could be partially predicted by the addition of phytoplankton metabolites[30]. However, the results from our split system (Supplementary Fig. 1h and Fig. 6), where bacterial SynComs formed distinct communities and had a beneficial effect on *Cr* growth depending on their physical proximity, indicate that the provision of diffusible carbon compounds is not sufficient to explain the observed patterns of microbial diversity. In addition, diffusibility gradients or macromolecular shedding (e.g., of *Cr* cell wall components), which may not be diffusible through the 0.22 µm-pore membrane, could also be involved. The importance of proximity to the algal cells could also be a consequence of gradients in concentrations and variations in the diffusivity of different compounds, which in aquatic environments are predicted to cause highly chemotactic, copiotrophic bacterial populations to outcompete low-motility oligotrophic ones[61]. Together with the algal growth data, the observed variations in SynCom structures suggest that, in addition to physical proximity, bi-directional exchange of metabolic currencies and/or molecular signals may be required for the assembly and sustained growth of a phycosphere microbiota capable of providing beneficial functions to their host. Future research will be aimed at exploring how green algae alter soil bacterial communities in terrestrial ecosystems, as well as elucidating molecular and ecological principles of microbiota establishment that might be shared with land plants.

## Methods

***Cr* culture conditions.** *Cr* CC1690 cells were grown photoautotrophically in TP[62], TP10, or B&D medium[63] at 25 °C, and the illumination of 125 µmol m$^{-2}$ s$^{-1}$ under continuous light conditions. Cultures were kept in a rotatory shaker at 70 RPM. Cells in the mid-logarithmic phase were used as inocula for the different experiments. Cell growth was determined either by measuring samples in a Multisizer 4e Coulter counter (Beckman Coulter Inc., California, USA) particle counter with the Beckman Coulter Multisizer software (v4.03) or using an Infinite M200Pro (TECAN Austria GmbH, Grödig, Austria) plate reader with the TECAN i-control software (v2.0.10.0), to determine either absorbance at 750 nm or chlorophyll fluorescence (excitation 440/9 nm, emission 680/20 nm).

**Isolation of subaerial microalgae.** Diverse subaerial algae (Supplementary Fig. 5), were isolated from a natural sample collected in July 2020 from a wet rock surface near Dreistegen, Monschau, Germany (50.550693N, 6.222094E). Enrichment of algae was achieved by the transfer of the natural sample into a general-purpose algal culture medium (modified Synthetic Freshwater Medium [SFM NH$_4$ pH6]; Supplementary Table 2). After several weeks under growth conditions (5 µmol photons m$^{-2}$ s$^{-1}$ with white LED illumination in a 14/10 h light/dark cycle at 20 °C), single cells/filaments were removed with a micropipette, washed several times in a sterile culture medium, and transferred to SFM NH$_4$ pH6 in 24-well microtiter plates. Single cell-derived, unialgal cultures have been deposited in the MEL algal research collection and are available upon request.

**Greenhouse experiment.** *A. thaliana* Col-0 seeds were surface-sterilized in 70% ethanol for 10 min followed by a brief wash with 100% ethanol (1 min), a wash with 3% NaClO (1 min), and five subsequent washes with sterile water. Seeds imbibed in sterile water were stratified for four days at 4 °C in the dark. Five seeds were then directly sown onto the surface of pots containing CAS by pipetting one seed at a time.

After 36 days plants were harvested similarly to previously reported protocols[21]. Briefly, plant roots were manually separated from the surrounding soil, until only tightly adhered soil particles were left. Then, roots were separated from their shoot and placed in a Falcon tube with 10 mL of deionized sterile water. After ten inversions, the roots were transferred to another Falcon tube and further processed, while leftover wash-off was centrifuged at 3,000 × *g* for 10 min. The supernatant was discarded and the pellet was resuspended and transferred to a new 2-mL screw-cap tube. This tube was centrifuged at 31,000 × *g* for 10 min, the supernatant was discarded, and the pellet snap-frozen in liquid nitrogen and stored for further processing (rhizosphere compartment). Root systems were then washed successively in 80% EtOH and 3% NaOCl to further clean the root surfaces from living microorganisms and subsequently washed three times (1 min each) in sterile water. These microbe-enriched root fractions were transferred to 2-mL screw-cap tubes for further processing (experiment A; Supplementary Fig. 1a and Supplementary Table 1).

Liquid TP cultures of 7-day old *Cr* (CC1690) were washed by sequential centrifugation at 1,900 × *g* for 5 min and resuspended in 50 mL of MgCl$_2$, to an average of 1.4 × 10$^6$ cells/mL across biological replicates, to be used as inocula for CAS pots. Samples from the surface of the *Cr*-inoculated pots were collected using an ethanol-washed metal spatula at 7, 14, 21, 28, and 36 days post-inoculation. Unplanted pots containing CAS were used to collect surface samples as mock-treatment control right after inoculation (day 0) and at the same time points as *Cr*-inoculated pots. The position of the pots in the trays was shuffled periodically to minimize edge and location effects. Sterile Petri dishes were placed at the bottom of each pot, which was then watered from the top at inoculation time with 50 mL of MgCl$_2$, and then by adding sterile MilliQ water every 2–3 days in the Petri dishes, and kept in the greenhouse under long-day conditions (16/8 h light/dark). Collected samples were snap-frozen using liquid nitrogen and stored at −80 °C until further processing.

To analyze the bacterial communities associated with native algae communities (experiment B; Supplementary Fig. 1b and Supplementary Table 1), CAS soil was placed in sterile "TP750 + TPD750" plastic boxes with filter lids (SacO2, Deinze, Belgium), at a depth of 3 cm, irrigated with sterile water, and incubated in a growth cabinet for 7 weeks with either a 12/12 h cycle at 19/21 °C. Negative controls were set up with a 0/24 h light/dark cycle and otherwise under the same conditions. At the end of the experiment, samples were obtained by harvesting the soil surface for DNA extraction and microbial community profiling. In addition, surface soil samples were obtained for chlorophyll extraction as a proxy for the growth of native and photosynthetically active algae.

**Microbial soil wash preparation.** Soil samples (5 g) from CAS or GOLM soil were collected in Falcon tubes and manually resuspended onto 30 mL of sterile 1× Tris-EDTA (TE) supplemented with 0.1% of Triton X-100 (SERVA Electrophoresis GmbH, Heidelberg, Germany). The solution was then homogenized by inversion at 40 RPM rpm for 30 min in a rotary mixer and centrifuged for 1 min at 450 ×*g* to remove bigger soil particles. Afterward, the supernatant was transferred to a new Falcon tube and centrifuged at 3,000 × *g* for 20 min. After centrifugation, the supernatant was discarded and the pellet resuspended in 50 mL of the final

medium. Cell concentration was then determined using either a hemocytometer or the Multisizer 4e.

**Mesocosm experiments.** *Cr* cells from an axenic culture were inoculated to a density of 10$^5$ cells/mL into 50 mL of TP or B&D medium in triplicate in 200 mL flasks. An estimate of 10$^9$ cells from the microbial soil wash was added to the same flasks and incubated for 11 days as described above. Controls consisted in flasks, wrapped in aluminum foil to prevent the pass of light, containing the same growth media as the one used for the *Cr* cultures with 0.1× AP[64] and without AP. Samples were collected for DNA extraction and cell counts determination at 0, 1, 4, 7, and 11 days post inoculation (experiment C, Supplementary Fig. 1c and Supplementary Table 1). These experiments were repeated in three biologically independent experiments, per soil type and growth media.

Subaerial algae (*Microthamnion* sp., MEL B 1108; *Chlamydomonas* sp. MEL 1030 B, *Spiroglea muscicola* MEL 1126 B, *Klebsormidium* sp. MEL1121 B, and *Chlamydomonas reinhardtii* CC1690) were inoculated to a similar chlorophyll fluorescence to that of the equivalent of 10$^5$ cells/mL for *Cr* into 50 mL of SFM medium without supplemented vitamins (Supplementary Table 2). An estimate of 10$^9$ cells from the microbial soil wash was added to the same flasks and incubated for 35 days under 12/12 h light/dark cycle without shaking. Samples were collected for DNA extraction and chlorophyll fluorescence determination at 0, 7, 14, 21, and 35 days post inoculation (experiment D; Supplementary Fig 1d and Supplementary Table 1).

**DNA extraction from soil samples.** Total DNA was extracted from the aforementioned samples using the FastDNA™ SPIN Kit for Soil following instructions from the manufacturer (MP Biomedicals, Solon, USA). DNA samples were eluted in 50 µL nuclease-free water and used for microbial community profiling.

**DNA extraction from liquid samples.** DNA from liquid samples was extracted using alkaline lysis[42]. Briefly, 12 µL of the sample were diluted in 20 µL of Buffer I (NaOH 25 mM, EDTA(Na) 0.2 mM, pH 12), mixed by pipetting, and incubated at 94 °C for 30 min. Next, 20 µL of Buffer II (Tris-HCl 40 mM, pH 7.46) was added to the mixture and stored at −20 °C.

**Chlorophyll extraction from soil samples.** Soil samples were oven-dried (60 °C) in 15 mL-falcon tubes for several days and chlorophyll extraction was performed by solvent extraction and shaking[65]. Briefly, a spatula tip of CaCO$_3$ (Carl Roth, Karlsruhe, Germany) was added to each soil sample as well as 6 mL of DMSO (Sigma-Aldrich, Darmstadt, Germany). Samples were boiled for 90 min in a water bath and then shaken for 20 min on a horizontal shaker. After placing the tubes in an upright position and allowing the soil to settle, the supernatant was transferred to a new tube and the procedure was repeated once more with the rest of the soil. Merged supernatants were used to measure chlorophyll fluorescence (excitation 440/9 nm, emission 680/20 nm).

**Isolation and genome sequencing of *Chlamydomonas*-associated bacteria.** Soil bacteria associated with *Cr* after co-cultivation were isolated from mesocosm cultures using a dilution-to-extinction approach[42,48]. Briefly, cultures containing *Cr* and bacteria from CAS soil washes as described above were incubated in TP or B&D media. After 7 days of co-cultivation, mesocosm samples were fractionated by sequential centrifugation and sonication[24] (Supplementary Fig. 1e) prior to dilution. For fractionation, cultures were centrifuged at 400 × *g* for 5 min to recover the supernatant. The pellet was washed with 1× TE buffer followed by sonication in a water bath at room temperature for 10 min and centrifugation at 1000 × *g* for 5 min. The supernatant from the first and second centrifugation was pooled together and diluted at either 1:10,000 or 1:50,000. Diluted supernatants were then distributed into 96-well microtiter plates containing 20% TSB media. After 3 weeks of incubation in the dark at room temperature, plates that showed visible bacterial growth were chosen for 16S rRNA amplicon sequencing. For identification of the bacterial isolates, a two-step barcoded polymerase chain reaction (PCR) protocol was used as previously described[48]. Briefly, DNA extracted from the isolates was used to amplify the v5–v7 fragments of the 16S rRNA gene by PCR using the primers 799F (AACMGGATTAGATACCCKG) and 1192R (ACGTCATCCCC ACCTTCC), followed by indexing of the PCR products using Illumina-barcoded primers. The indexed 16S rRNA amplicons were subsequently pooled, purified, and sequenced on the Illumina MiSeq platform. Next, cross-referencing of IPL sequences with mesocosm profiles allowed us to identify candidate strains for further characterization, purification, and whole-genome sequencing. Two main criteria were used for this selection: first, we aimed at obtaining maximum taxonomic coverage and selected candidates from as many taxa as possible; second, we gave priority to strains whose 16S sequences were highly abundant in the natural communities. Whenever multiple candidates from the same phylogroup were identified, we aimed at obtaining multiple independent strains, if possible, coming from separate biological replicates to ensure they represented independent isolation events. After validation of selected strains, 185 were successfully subjected to whole-genome sequencing. Liquid cultures or swabs from agar plates from selected bacterial strains (Supplementary Data 3) were used to extract DNA using the QiAmp Micro DNA kit (Qiagen, Hilden, Germany). The extracted DNA was

treated with RNase and purified. Quality control, library preparation, and sequencing (2 ×150 bp; Illumina HiSeq3000) at a 4–5 million reads per sample were performed by the Max Planck-Genome Centre, Cologne, Germany (https://mpgc.mpipz.mpg.de/home/).

**Multi-species microbiota reconstitution experiments**. The gnotobiotic FlowPot[66] system was used to grow Cr or A. thaliana plants with and without bacterial SynComs (experiment G; Supplementary Fig. 1g and Supplementary Table 1). This system allows for even inoculation of each FlowPot with microbes by flushing the pots with the help of a syringe attached to the bottom opening. After FlowPot assemblage, sterilization and microbial inoculation sterilized seeds were placed on the matrix (peat and vermiculite, 2:1 ratio), and pots were incubated under short-day conditions (10 h light, 21 °C; 14 h dark, 19 °C), standing in customized plastic racks in sterile "TP1600 + TPD1200" plastic boxes with filter lids (SacO2, Deinze, Belgium). For SynCom preparation, bacterial strains from either Cr- or At-SPHERE were grown separately in liquid culture for 2–5 days in 50% TSB media and then centrifuged at 4000×g for 10 min and re-suspended in 10 mM MgCl$_2$ to remove residual media and bacteria-derived metabolites. Equivalent ratios of each strain, determined by optical density (OD600) were combined to yield the desired SynComs (Supplementary Data 4). An aliquot of the SynComs as reference samples for the experiment microbial inputs was stored at −80 °C for further processing. SynCom bacterial cells (10$^7$) were added to either 50 mL of TP10 or ½ MS (Duchefa Biochemie, Haarlem, Netherlands), which were then inoculated into the FlowPots using a 60 mL syringe. For Cr-inoculated pots, 10$^5$ washed Cr cells were added to the 50 mL of media with or without microbes to be inoculated into the FlowPots.

Chlamydomonas or Arabidopsis FlowPots were grown side-by-side in gnotobiotic boxes, with six pots in total per box. This experiment was repeated in three independent biological replicates. After five weeks of growth, roots were harvested and cleaned thoroughly from attached soil using sterile water and forceps. The surface of Chlamydomonas pots was used as phycosphere samples (cells were harvested from visibly green surface areas, topsoil samples). In addition, to remove any possible background effect from carry-over soil particles, the surface-harvested samples were washed in sterile TE supplemented with 0.1% of Triton X-100 by manually shaking in 2-mL Eppendorf tubes. Then, the tubes rested for a few minutes and the supernatant was used as "cell fraction" samples. Finally, soil from unplanted pots was collected as soil samples and treated similarly to Chlamydomonas-inoculated pots for microbial community comparison. All phycosphere, root (comprising both the epiphytic, and endophytic compartments), and soil (soil from unplanted pots) samples were transferred to Lysing Matrix E tubes (MP Biomedicals, Solon, USA), frozen in liquid nitrogen, and stored at −80 °C for further processing. DNA was isolated from those samples using the MP Biomedicals FastDNA™ Spin Kit for Soil, and from the input SynCom by alkaline lysis, and subjected to bacterial community profiling.

To ensure sufficient surface for phycosphere harvesting, we set up an additional experiment based on sterile peat without FlowPots. Experiments with the mixed SynCom of Cr- and At-SPHERE strains were conducted using sterile "TP750 + TPD750" plastic boxes (SacO2, Deinze, Belgium). Sterile soil and vermiculite were mixed in a 2:1 ratio and added to each box. Next, the boxes were inoculated by adding 95 mL of TP10 or ½ MS, for the Chlamydomonas or Arabidopsis boxes respectively, containing 2 × 10$^7$ bacterial cells.

Samples for chlorophyll extraction were collected from the different Chlamydomonas containing gnotobiotic systems by harvesting the green surface of the peat and extracting the cells as described above. Then, 1 mL of these extracts were centrifuged at 3,000 × g for 1 min at 4 °C with 2.5 µl 2% (v/v) Tween 20 (Sigma-Aldrich, Darmstadt, Germany) to promote the aggregation into a pellet. Then, the supernatant was completely removed and the pellets stored at −80 °C until extraction.

**Chlorophyll extraction from algae-containing samples**. From each extracted cell sample from the gnotobiotic soil system, 1 mL was collected and mixed with 2.5 µL of 2% (v/v) Tween 20 in 1.5 mL Eppendorf tubes. The samples were centrifuged for 1 min at 3,000 × g and 4 °C, then the supernatant was removed and the pellet stored at −80 °C. Frozen samples were thawed on ice for 2 min and 1 mL of HPLC grade methanol (Sigma, 34860-4L-R) was added to the pellets. The tubes were covered from the light using aluminum foil and mixed using the vortex for 1 min. After vortexing, the cells were incubated in the dark at 4 °C for 5 min. Next, the pigments were obtained by centrifuging the cells for 5 min at 3,000 × g and 4 °C and recovering the supernatant. The pigments absorbance at 652 and 665 nm was measured in a plate reader Infinite M200Pro using methanol as blank. The absorbance values were then substituted in the following equation Chl a + Chl b = 22.12 × Abs$_{652}$ + 2.71 × Abs$_{665}$[67].

**Split co-cultivation system**. Co-cultivation devices were built by adapting 150 mL Stericup-GV filtration devices (Merck Millipore, Darmstadt, Germany) harboring a 0.22 µm filter membrane[68]. Each co-cultivation device was assembled inside a clean hood 150 mL and 100 mL of TP10 were added into the big and small chamber of the filtration device, respectively. Chambers were inoculated at different cell concentrations depending on the content of the chamber (experiment H;

Supplementary Fig 1h and Supplementary Table 1). The concentrations used were 10$^5$ and 10$^7$ cells/mL for Chlamydomonas and SynCom respectively. For the C + SC condition, the inoculum concentration was the same as for individual content chambers. After inoculation, the devices were transferred to a shaking platform and incubated under the same conditions used for Cr liquid cultures described above. Four samples per chamber were harvested for DNA extraction, fluorescence, and cell growth at the start of the incubation and 7 days after inoculation. These experiments were repeated in three independent biological replicates, containing one technical replicates each.

Additionally, a full-factorial replicate of the experiment (experiment H; Supplementary Fig. 1h and Supplementary Table 1) was carried out using a custom-made co-cultivation device (Cat. #0250 045 25, WLB Laborbedarf, Möckmühl, Germany). Briefly, two 250 mL borosilicate glass bottles were modified by adding on the sidewall of each bottle a glass neck with an NW25 flange. The flange holds a disposable 0.22 µm-pore PVDF Durapore filtration membrane (Merck Millipore, Darmstadt, Germany) and is kept in place by an adjustable metal clamp. In this device, each bottle holds 150 mL of TP10, and the initial cell concentrations were the same as the ones used in the previously described co-cultivation device. Similar to the Stericup system, four samples per chamber were harvested for DNA extraction. Chlorophyll fluorescence and cell growth measurements were collected at the start of the incubation and 7 days after inoculation. These experiments were repeated in three independent biological replicates, containing one technical replicates each.

**Preparation of SynCom inocula**. Bacterial cultures from the strains selected for the different SynComs (Supplementary Data 4) were started from glycerol stocks which were used to streak agar plates containing TSA 50% media. Plates were cultured at 25 °C for five days and later used to inoculate culture tubes with 1 mL of 50% TSB media. The tubes were incubated for six days at 25 °C and 180 RPM. After 6 days, the cultures were washed three times by centrifugation at 3,000 × g for 5 min, the supernatant discarded, and the pellet resuspended into 2 mL of TP or TP10 media. The washed cultures were further incubated with shaking at 25 °C for an additional day. Bacterial concentration in washed cultures was determined by measuring OD$_{600}$ and, subsequently pooled in equal ratios. Cell counts of the pooled SynCom were measured using the Multisizer 4e and adjusted to 10$^6$, to inoculate together with 10$^4$ cells of Cr (prepared as described above) in 50 mL of TP10 in 200-mL flasks. These flasks were inoculated in triplicate and three biological replicates were prepared for both bacteria and Cr start inocula. As controls, Cr-only cultures and SynCom-only cultures were incubated in parallel, and samples were taken at 0, 1, 4, 7 for community profiling, and at 0, 4, 7, 14 days for Cr cell counts.

**Culture-independent bacterial 16S and 18S rRNA sequencing**. DNA samples were used in a two-step PCR amplification protocol. In the first step, V2–V4 (341F: CCTACGGGNGGCWGCAG; 806R: GGACTACHVGGGTWTCTAAT), V4-V7 (799F: AACMGGATTAGATACCCKG; 1192R: ACGTCATCCCCACCTTCC) of the bacterial 16S rRNA, or V9 (F1422: ATAACAGGTCTGTGATGCCC; R1797: TGATCCTTCTGCAGGTTCACCTAC) of the eukaryotic 18S rRNA were amplified. Under a sterile hood, each sample was amplified in triplicate in a 25 µL reaction volume containing 2 U DFS-Taq DNA polymerase, 1× incomplete buffer (Bioron GmbH, Ludwigshafen, Germany), 2 mM MgCl$_2$, 0.3% BSA, 0.2 mM dNTPs (Life technologies GmbH, Darmstadt, Germany) and 0.3 µM forward and reverse primers. PCR was performed using the same parameters for all primer pairs (94 °C/2 min, 94 °C/30 s, 55 °C/30 s, 72 °C/30 s, 72 °C/10 min for 25 cycles). Afterwards, single-stranded DNA and proteins were digested by adding 1 µL of Antarctic phosphatase, 1 µL Exonuclease I, and 2.44 µL Antarctic Phosphatase buffer (New England BioLabs GmbH, Frankfurt, Germany) to 20 µl of the pooled PCR product. Samples were incubated at 37 °C for 30 min and enzymes were deactivated at 85 °C for 15 min. Samples were centrifuged for 10 min at 3,000 × g and 3 µl of this reaction were used for a second PCR, prepared in the same way as described above using the same protocol but with cycles reduced to 10 and with primers including barcodes and Illumina adapters. PCR quality was controlled by loading 5 µL of each reaction on a 1% agarose gel and affirming that no band was detected within the negative control. A were loaded on a 1.5% agarose gel and run for 2 h at 80 V. Subsequently, bands with a size of ~500 bp were cut out and purified using the QIAquick gel extraction kit (Qiagen, Hilden, Germany). DNA concentration was determined from fluorescently measurements, and 30 ng DNA of each of the barcoded amplicons were pooled in one library. The library was then purified and re-concentrated twice with Agencourt AMPure XP beads, and pooled in similar ratios for sequencing. Paired-end Illumina sequencing was performed in-house using the MiSeq sequencer and custom sequencing primers.

**Analysis of culture-independent bacterial 16 S rRNA profiling**. Amplicon sequencing data from Cr or At roots grown in CAS soil in the greenhouse, along with unplanted controls, were demultiplexed according to their barcode sequence using the QIIME pipeline[69]. Afterward, DADA2[70] (v1.16) was used to process the raw sequencing reads of each sample. Unique amplicon sequencing variants (ASVs) were inferred from error-corrected reads, followed by chimera filtering, also using the DADA2 pipeline. Next, ASVs were aligned to the SILVA database[71]

(v138) for a taxonomic assignment using the naïve Bayesian classifier implemented by DADA2. Raw reads were mapped to the inferred ASVs to generate an abundance table, which was subsequently employed for analyses of diversity and differential abundance using R (v4.0.3) and the R package *vegan*[72](v2.5–6).

Amplicon sequencing reads from the *Cr* IPL and from the corresponding mesocosm culture-independent community profiling were quality-filtered and demultiplexed according to their two-barcode (well and plate) identifiers using custom scripts and a combination of tools included in the QIIME (v1.9.1) and USEARCH (v8.0.1517)[73] pipelines. Next, sequences were clustered into OTUs with a 97% sequence identity similarity using the UPARSE algorithm, followed by identification of chimeras using UCHIME[74,75]. We performed this analysis at the OTU level to be able to directly compare with earlier data from[42], where the *A. thaliana* culture collection was described. Samples from wells with fewer than 100 good quality reads were removed from the data set as well as OTUs were not found in a well with at least ten reads. Recovery rates (Supplementary Fig. 6a). were estimated by calculating the percentage of the top 100 most abundant OTUs found in natural communities (greenhouse experiment) that had at least one isolate in the culture collection (62%), and the total aggregated relative abundances of recovered OTUs (63%). We identified IPL samples matching OTUs found in the culture-independent root samples and selected a set of 185 representative strains maximizing taxonomic coverage for subsequent validation and whole-genome sequencing, forming the basis of the *Cr*-SPHERE collection.

Finally, to test whether the observed overlap between the *At* root and *Cr* phycosphere communities could be explained by a random process of recruitment from the same starting soil microbiota using the hypergeometric distribution. Briefly, we assumed that abundant members of the *Cr* phycosphere (141 ASVs) were randomly drawn from the starting soil community (3537 ASVs) and then tested whether the observed overlap (43 out of 120 ASVs) could be explained by chance.

**Analysis of culture-independent eukaryotic 18S rRNA profiling**. Amplicon sequencing data from soil surface samples collected at the start and end of the soil native algae community experiment (experiment B, Supplementary Fig. 1b and Supplementary Table 1) were demultiplexed according to their barcode sequence using the QIIME pipeline[69] (v1.9.1). Afterward, DADA2[70] (v1.16) was used to process the raw sequencing reads of each sample. Unique ASVs were inferred from error-corrected reads, followed by chimera filtering, also using the DADA2 pipeline. Next, ASVs were aligned to the Protist Ribosomal Reference (PR2 v4.14.0) database[76] for a taxonomic assignment using the 'assign_taxonomy.py' script from QIIME. Raw reads were mapped to the inferred ASVs to generate an abundance table, which was subsequently employed for analyses of diversity and differential abundance in R (v4.0.3) using the R package *vegan*[72] (v2.5–6).

**Meta-analysis of phycosphere and root microbiota profiles**. For the meta-analysis of root microbiota samples across plant species, data from previous studies of *Arabidopsis* and *Lotus* grown in CAS soil[7,43] (ENA accessions: PRJEB38663, and PRJEB34100, respectively) were processed using the pipeline described above and merged with samples obtained from the Cooloola natural site chronosequence[11]. Sequencing reads from this latter study (ENA accession: PRJNA328519; Roche 454) were quality filtered and trimmed after the removal of primer sequences. Given that these studies employed non-overlapping sequencing primers, all datasets were combined after aggregating relative abundances at the bacterial order taxonomic level. The core taxa of the root microbiota were determined by identifying bacterial orders present in every plant species with an occupancy of at least 80% (i.e., found in at least 80% of the root samples of a given species) with a relative abundance above 0.1%. To infer the phylogenetic relationship between the different hosts, protein sequences of the ribulose-bisphosphate carboxylase (*rbcL*) gene for *Cr* and the 35 analyzed plant species were recovered from GenBank. The sequences were aligned using Clustal Omega[77] (v1.2.0) with default parameters, and the alignment was used to infer a maximum likelihood phylogeny using FastTree[78]. (v2.1.3).

**Analysis of culture-dependent amplicon sequencing data**. Sequencing data from SynCom experiments was pre-processed similarly as natural community 16S rRNA data. Quality-filtered merged paired-end reads were then aligned to a reference set of sequences extracted from the whole-genome assemblies of every strain included in a given SynCom, using Rbec[79] (v1.0.0). We then checked that the fraction of unmapped reads did not significantly differ between compartment, experiment, or host species. Next, we generated a count table that was employed for downstream analyses of diversity in R (v4.0.3) with the R package vegan (v2.5–6). Finally, we visualized amplicon data from all experimental systems using the *ggplot2* R package[80] (v3.3.2).

**Genome assembly and annotation**. Paired-end Illumina reads were first trimmed and quality-filtered using Trimmomatic[81](v0.32). QC reads were assembled using the IDBA (v1.1.3) assembler[82] within the A5 pipeline[83] (v20160825). Assembly statistics and metadata from the assembled genomes can be found in Supplementary Data 3. Genome assemblies with either multi-modal *k*-mer and G + C content distributions or multiple cases of marker genes from diverse taxonomic

groups were flagged as not originating from clonal cultures. Such assemblies were then processed using a metagenome binning approach[84]. Briefly, contigs from each of these samples were clustered using METABAT2[85] (v2.2.13) to obtain metagenome-assembled genomes (MAGs). Each MAG was analyzed to assess completeness and contamination using CheckM[86] (v1.1.2). Only bins with completeness scores larger than 75% and contamination rates lower than 5% were retained and added to the collection (Supplementary Data 3; designated MAG in the column "type"). Classification of the bacterial genomes into phylogroups was performed by calculating pair-wise average nucleotide identities using FastANI[87] (v1.32) and clustering at a 97% similarity threshold. Functional annotation of the genomes was conducted using Prokka[88] (v1.12-beta) with a custom database based on KEGG Orthologue (KO) groups[50] downloaded from the KEGG FTP server in November 2019. Hits to sequences in the database were filtered using an *E*-value threshold of $10 \times 10^{-9}$ and minimum coverage of 80% of the length of the query sequence.

**Comparative genome analyses of the *Cr*-, *At*- and *Lj*-SPHERE culture collections**. The genomes from the *Cr*-, *At*-, and *Lj*-SPHERE culture collections[42,48] were queried for the presence of 31 conserved, single-copy marker genes, known as AMPHORA genes[89]. Next, sequences of each gene were aligned using Clustal Omega[77] (v1.2.0) with default parameters. Using a concatenated alignment of each gene, we inferred a maximum likelihood phylogeny using FastTree[78] (v2.1.3). This tree was visualized using the Interactive Tree of Life web tool[90]. Finally, genomes from the three collections (*Cr*-SPHERE, *At*-SPHERE, and *Lj*-SPHERE) were clustered into phylogroups, roughly corresponding to a species designation[91] using FastANI[87] (v1.32) and a threshold of average nucleotide identity at the whole genome level of at least 97%. Functional comparison among the genomes from the *Cr*-, *Lj*-, and *At*-SPHERE collections were performed by comparing their annotations. KO groups were gathered from the genome annotations and aggregated into a single table. Lastly, functional distances between genomes based on Pearson correlations were used for principal coordinate analysis using the *cmdscale* function in R (v4.0.3).

**Statistics and reproducibility**. All experiments were performed with full factorial (biological and technical) replication. In each case, the number of samples (*n*), distributed across at least three independent biological replicates, is indicated in the corresponding figure captions. Whenever microbial abundances or plant growth parameters were compared, we used a two-sided, non-parametric Mann–Whitney test or, in the case of multiple comparisons, a Kruskal–Wallis test, followed by a Dunn's post hoc. When appropriate, *P* values were adjusted for multiple testing using the Benjamini–Hochberg method ($\alpha = 0.05$). Statistical tests on beta-diversity analyses were performed using a PERMANOVA test with 5000 random permutations. Whenever boxplots were used in figures, data were represented as median values (horizontal line), Q1 – 1.5× interquartile range (boxes), and Q3 + 1.5× interquartile range (whiskers).

**Reporting summary**. Further information on research design is available in the Nature Research Reporting Summary linked to this article.

## Data availability
Raw sequencing data have been deposited into the European Nucleotide Archive (ENA) under the accession number PRJEB43117. Source data are also provided with this paper. In addition, sequencing data, intermediate results, and metadata tables can be downloaded as a bundle from the *At*-SPHERE website (http://www.at-sphere.com/cr.tar.gz). Source data are provided with this paper.

## Code availability
The scripts used for the computational analyses described in this study are available at GitHub (http://www.github.com/garridoo/crsphere; https://doi.org/10.5281/zenodo.5767307) to ensure the replicability and reproducibility of these results.

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

## Acknowledgements

We would like to thank Dr. Paul Schulze-Lefert for his support and advice throughout the duration of this project. We would also like to thank Dr. Stephane Hacquard, Dr. Thomas Nakano, Dr. Oliver Ebenhöh, and Dr. Andreas Weber for their critical comments on this manuscript. We thank Dr. Ralph Bock, Dr. Juliane Neupert, and Dr. Ru Zhang for their advice and assistance during the early stages of this research project. Finally, we thank Rozina Kardarakis for the grammatical and style corrections of the manuscript. This research was funded by the Max Planck Society and the Deutsche Forschungsgemeinschaft (DFG, German Research Foundation) under Germany's Excellence Strategy—EXC-Nummer 2048/1—project 390686111, and the "2125 DECRyPT" Priority Program.

## Author contributions

P.D., J.F.-U., K.W., B.M., M.M., and R.G.-O. designed the experiments. P.D. and J.F.-U. conducted the greenhouse and soil native algae community experiments. P.D. and K.W. performed the mesocosm experiments. P.D., J.F.-U., and K.W. established the IPL bacterial library and characterized the *Cr*-SPHERE core culture collection. B.M. and M.M. isolated the environmental algal strains. P.D. conducted the experiment with the environmental algal strains. P.D. and J.F.-U. performed the synthetic community experiments. P.Z., J.F.-U., and R.G.-O. analyzed whole-genome sequencing data. P.D., J.F.-U., P.Z., R.G., and R.G.-O. analyzed the amplicon data. P.D., J.F.-U., K.W., B.M., M.M., and R.G.-O interpreted the results. P.D., J.F.-U., and R.G.-O wrote the paper.

## Funding

## Competing interests

The authors declare no competing interests.
