## [Peer Review File · Nature Communications]

Shared features and reciprocal complementation of the Chlamydomonas and Arabidopsis microbiotaREVIEWER COMMENTS

Reviewer #1 (Remarks to the Author):

This manuscript describes the bacterial communities found associated with *Arabidopsis* (At) roots and *Chlamydomonas* (Cr) cultures. Sequence data from 16S metabarcoding identified bacterial taxa at the phylum level to be compared and similarities were found. Subsequent experiments indicated that both eukaryotic hosts were able to 'assemble' taxonomically equivalent microbiomes from a larger sample and that synthetic communities made up of representative species from the At microbiome were able to support growth of Cr, and vice versa. Whole genome sequence data was also obtained from 185 bacterial isolates and whole genome phylogenies were generated, again confirming that the major groups of bacteria associated with At were also found in Cr. The authors conclude that common organisational principles for assembling microbiomes are shared between chlorophytes and land plants. The study is well executed and the data appear of good quality and statistically significant, so the conclusion that At roots and Cr assemble very similar groups of bacteria is robust. The work will be of interest to microbial ecologists particularly those working with soil communities.

Specific points

1. However, the motivation behind the study and the rationale for the various experiments is not clear. The Introduction starts by describing the plant microbiota and how these have been studied across the taxonomic groups from liverworts to angiosperms, and then proposes that there may be similarities with the bacterial communities that associate with algae. But is this because of evolutionarily conserved mechanisms or is it merely an obvious consequence of bacteria that can utilise the products of photosynthesis? The overlap between the bacterial communities are correlations but without the equivalent of an outgroup, this does not demonstrate causality. Particularly because only one algal species was used, this cannot be extrapolated to a specific mechanistic process of bacterial-host interactions
2. The choice of *Chlamydomonas* as the alga to compare with *Arabidopsis* (and other land plants) is not explained - it is indeed a soil alga but the CC1690 strain chosen for the study is a laboratory strain that has been maintained in axenic culture for decades and so may well not be representative of environmental strains that are adapted to living within complex communities. Cr is often used for a range of studies for biotic interactions - but in these cases the justification is that it is easy to manipulate with sequenced and well-annotated genome, and provisos are made about extrapolating to the environment.
3. More importantly, Cr is not related to the charophyte algae from which land plants evolved (see for example Delwiche & Cooper 2015 *Curr Biol* 25(19):R899-910. doi: 10.1016/j.cub.2015.08.029) and so there is little reason to suppose that there are evolutionarily conserved mechanisms in terms of interactions with bacteria, other than between eukaryotes generally. On lines 105-108, the authors describe the selection of bacterial taxa representative of 'phylogenetically diverse plant species' - and yet these are compared with just one algal species that is likely far more distantly related to charophytes than lycopods are to angiosperms. Throughout the text there are statements that imply that Cr CC1690 is representative of green algae (eg lines 25-26, 117-118, 301, 317-318, 333-334, among others), providing a misleading impression. Indeed the text on lines 117-118 extrapolates Cr as representing a wide range of photosynthetic unicellular eukaryotes, belying the diversity across the eukaryotic tree of life.
4. I was not clear about the purpose of whole genome sequencing work, since it shows similar results to the 16S metabarcoding. As mentioned above, the correlations between the composition of the microbiota may simply be just that. The WGS data might have provided the means to assess metabolic capabilities between those taxa that were found associated compared to the soil bacteria generally but there was no attempt to do that. Incidentally the ENA accession number does not provide access to the sequence data.
5. Lines 307-310 - it is not just cell wall components that might be present in the phycosphere. Cr secretes a range of fermentation products at night including formate, acetate and H₂.

6. lines 141-143 - it is not clear what this experiment is showing - which fraction is being measured?

Reviewer #2 (Remarks to the Author):

The paper is well written and undoubtedly shows that *Chlamydomonas* can alter a bacteria community whether in soil or in a syncom. However, I don't really know what this means as adding a photosynthetic eukaryote to any bacterial community will result in changes to the community. I also accept that while diffusion of carbon and probably other metabolites from the algae is important there are proximity effects i.e. when the algae are present with they community they shift is structure (albeit very slightly Figure 5c). However, diffusion gradients and macromolecular shedding might cause similar effects. I also accept that the communities assembled by the algae and plants have similarities although also big differences. However, again that is hardly surprising as the same soil community is the ultimate source of the bacteria. A much more difficult question is whether the algae really alters the bacterial community in a meaningful way in a natural setting and whether there is any mechanistic interaction between the algae and the bacterial community. This is clear with plant roots where it can be shown that bacteria attach to roots and form biofilms. Flooding soil with *Chlamydomonas* and measuring changes does not look like a natural situation to me.

minor points

- The use of symbol sizes in figure 2 and support fig 4 is pretty much impossible to distinguish in the figure
- The legend in figure three appears to be scrambled, panel a is really c, b is d, c is a and d is b.

Reviewer #3 (Remarks to the Author):

In this manuscript, Durán, Flores-Urbe et al. describe the microbiota of the algae *Chlamydomonas* and compare its taxonomic composition and genomic features to the plant root microbiota. The authors use publicly available data on the microbiota of a variety of plants, in addition their own data from previous studies on the model plant *Arabidopsis*, and identify core taxa that are present in all hosts. Moreover, the authors establish an extensive culture collection of *Chlamydomonas*-associated bacteria and perform synthetic community (SynCom) experiments. In this study, the authors performed experiments with single SynComs and competition between SynComs of *Chlamydomonas* and *Arabidopsis*. Depending on the experiments, the SynCom consisted of up to 26 strains. The SynComs are designed to contain equivalent taxa (9 strains from each host), and were able to assemble communities with similar composition on both hosts. During competition of both SynComs, strains performed better in their "native" host compared to strains isolated from the other host. To explore whether the release of diffusible photoassimilates from the algae could explain community composition, the authors performed split co-culture experiments and analyze algal growth and community composition. The communities were different when *Chlamydomonas* and bacteria could only exchange metabolites compared to when both were in physical contact to each other.

Overall, the manuscript is well written and the figures are beautifully crafted. The study is original, and of particular interest to the plant and algae research communities. *Chlamydomonas* is a model organism and its research in this context fosters our understanding of host-associated microbiota assembly. Foremost, the new culture collection provides a valuable tool for future studies. The experimental procedures are sound and the authors have a long-standing experience in 16S rRNA amplicon data analysis. The experiments are well designed and give important insights in microbiota assembly.

Still, I have some comments that would improve the manuscript:

1) I wonder what defines "the phycosphere community" when the authors summarize the results of the split co-culture experiment? The authors show differences in community compositions but no details how the communities differ. Surely, the communities are different and physical contact has an additional influence, but how does this difference compare to the difference of phycosphere community composition between soil, liquid shake flask and with *At* strains? I would suggest that the authors take this into account when they seemingly talk about "the" phycosphere community (e.g. line 23) and maybe rephrase.

2) Similarly, I stumbled over the "core" ecological principles (line 24) as being a very vague and general expression, especially since no further explanation is given what it actually means in the specific context of the study. Do the authors mean that the "core" (or basic) principle is provision of nutrients/metabolites by the host in general? Or that similar nutrients are provided by *Arabidopsis* and *Chlamydomonas*? Or that nutrients and contact are required for a fully differentiated microbiota assembly in *Arabidopsis* and *Chlamydomonas*? Please clarify.

3) I have a question regarding Supp. Fig. 2b-d and Fig.1c. Theoretically, the same ASV could be enriched in the root and phycosphere. Looking at Supp. Fig. 2b-d, this does not seem to be the case (or only for very low abundant strains). However, there seem to be some commonality between root and phycosphere (Fig1c). Is this similarity only seen at OTU level as presented in Fig.1c? Maybe the authors could explain in the text why they switched the taxonomic level for different analysis.

4) In Fig.1d, I was surprised to see Firmicutes clustering with the Alphaproteobacteria in the phylogenetic tree based on 16S sequences. It seems to be different when using AMPHORA genes. I guess it is correct, but it was just unexpected to me.

5) Comment: Regarding the experiments of Supp. Fig4b and Fig.5, a potential future experiment could be to mimic physical contact by heat killed *Chlamydomonas* and adding the Artificial Photosynthates or using the split co-culture.

6) I appreciate the time-course experiments to monitor community assembly. In Fig.3d, why did the authors include the rel. abundance of *Chlamydomonas* in the graph? While it is interesting to see that *Chlamydomonas* abundance is as high as entire microbiota, the community composition of Input, SynCom alone and Cr+SynCom cannot be compared anymore.

7) Fig. 4cd show the aggregated relative abundance. As this is the average of all strains, I wonder whether this is mainly driven by few strains or if most strains show such a host-specificity?

8) In multiple experiments, the authors use *Chlamydomonas* fluorescence to assess its abundance (e.g. Fig5d, Fig3b). As bacteria can also release fluorescent compounds (e.g. siderophores), I wonder whether it is a suitable method to monitor changes in Cr abundance when comparing presence and absence of a SynCom. Measuring the fluorescence of a bacterial supernatant filtrate (+/- *Chlamydomonas*) would reveal if there is a background due to secretion of bacterial compounds.

9) Throughout the manuscript, the figure legends mostly interpret the results and only provide minimal technical information. Personally, I rather prefer if the legends states the experimental settings, sample number, statistics etc. to make it possible to understand how the data shown was generated. Often, the method section does not cover every experimental detail or small variations of the specific experiment shown in the respective figure.

10) In the discussion of the data presented in Figure 4 and 5, the authors follow the model of sequential selection of microbiota members by the host that gets increasingly specific. Considering the "reciprocal complementation" of both SynComs on family level, the host (or ecological factors) applies a rough filter to select on a "high taxonomic level". Only in direct competition between close relatives, strain-specific adaptations to their respective host become observable. I totally agree with this argumentation, but I got a bit confused because first the authors highlight differences between extracellular organic carbon released from *Arabidopsis* and *Chlamydomonas* (307 f) and later suggest the release of photoassimilates as first principle driving community

assembly in these hosts (339 f). I would assume a comparison of exudates from both organism could be insightful to clarify these questions.

Minor points:

- 1) The At and Cr SynComs "assemble into taxonomically equivalent communities". If the input is taxonomically equivalent, all strains will be detected in the end. Maybe mention that the relative abundance/composition rather taxonomy is similar.
- 2) Fig.3: the description in the legend does not correspond to the correct panels.
- 3) As some beta-diversity analysis are done on ASV, OTU and family-level, it would be important to mention in each figure legend what level was analysed (e.g. Fig5a).

We would like to thank all three reviewers for their helpful comments and their critical evaluation of our work. In response to the issues raised by the reviewers we have performed several additional experiments, which we outline below:

1. Culture-independent characterization of the phycosphere microbiota of a selection of subaerial algal strains, isolated from the same terrestrial environment, and which contain representatives from phylogenetically diverse algae, including multiple chlorophyte and streptophyte algal isolates. The results of this survey show that assembly of distinct phycosphere bacterial communities, enriched in bacterial taxonomic groups shared with the root microbiota, can be found in representatives from diverse lineages of green algae.
2. An experiment using natural soil without *C. reinhardtii* inoculation, where we allowed a community of native microalgae to develop without external intervention. The results from this experiment show that naturally-occurring phycosphere communities from terrestrial ecosystems resemble the *C. reinhardtii*-associated communities characterized in our previous experiments.
3. A computational analysis in which we simulated the process of recruitment of bacterial species from the starting soil microbiota by *C. reinhardtii* using the hypergeometric distribution. The results from this test demonstrate that the overlap between the *Cr* phycosphere and *At* root communities is highly significant ($P=3.06\times 10^{-33}$) and cannot be explained by a random process of microbial recruitment from the same starting soil community.

We believe that these new data support and complement our earlier findings and allow us to extend our main conclusions to representative members of other algal taxonomic groups, some of which are more closely related to extant embryophytes than to chlorophyte algae such as *C. reinhardtii*. Below, we provide a description of these new findings and a detailed point-by-point response to the reviewers' comments.

Reviewer #1 (Remarks to the Author):

This manuscript describes the bacterial communities found associated with *Arabidopsis* (*At*) roots and *Chlamydomonas* (*Cr*) cultures. Sequence data from 16S metabarcoding identified bacterial taxa at the phylum level to be compared and similarities were found. Subsequent experiments indicated that both eukaryotic hosts were able to 'assemble' taxonomically equivalent microbiomes from a larger sample and that synthetic communities made up of representative species from the *At* microbiome were able to support growth of *Cr*, and vice versa. Whole genome sequence data was also obtained from 185 bacterial isolates and whole genome phylogenies were generated, again confirming that the major groups of bacteria associated with *At* were also found in *Cr*. The authors conclude that common organisational principles for assembling microbiomes are shared between chlorophytes and land plants. The study is well executed and the data appear of good quality and statistically significant, so the

conclusion that *At* roots and *Cr* assemble very similar groups of bacteria is robust. The work will be of interest to microbial ecologists particularly those working with soil communities.

We thank the reviewer for their positive evaluation of our work and their constructive comments.

Specific points

1.1. However, the motivation behind the study and the rationale for the various experiments is not clear. The Introduction starts by describing the plant microbiota and how these have been studied across the taxonomic groups from liverworts to angiosperms, and then proposes that there may be similarities with the bacterial communities that associate with algae. But is this because of evolutionarily conserved mechanisms or is it merely an obvious consequence of bacteria that can utilise the products of photosynthesis? The overlap between the bacterial communities are correlations but without the equivalent of an outgroup, this does not demonstrate causality. Particularly because only one algal species was used, this cannot be extrapolated to a specific mechanistic process of bacterial-host interactions

We agree with the reviewer that extrapolating our results with *C. reinhardtii* to a general mechanism of interactions between photosynthetic hosts and heterotrophic bacteria was not justified, particularly given that only one algal species had been used. To address this limitation, we have performed a new experiment where we characterized the phycosphere communities of a panel of diverse subaerial chlorophyte and streptophyte algae (please, see also our response to point 1.3. below).

In addition, we have now extensively modified the introduction of our manuscript to more clearly explain the motivation behind the study and the rationale for the various experiments.

1.2. The choice of *Chlamydomonas* as the alga to compare with *Arabidopsis* (and other land plants) is not explained - it is indeed a soil alga but the CC1690 strain chosen for the study is a laboratory strain that has been maintained in axenic culture for decades and so may well not be representative of environmental strains that are adapted to living within complex communities. *Cr* is often used for a range of studies for biotic interactions - but in these cases the justification is that it is easy to manipulate with sequenced and well-annotated genome, and provisos are made about extrapolating to the environment.

We agree with the reviewer that a laboratory strain that has been maintained in axenic culture for decades might not be representative of environmental strains that are adapted to living within complex soil bacterial communities. To address this potential concern, we have now performed an additional experiment, in which we directly compare the *C. reinhardtii* laboratory strain CC1690 with a *Chlamydomonas* sp. isolate from a terrestrial environment. The results from this experiment show that these two *Chlamydomonas* strains assemble similar phycosphere communities (see new **Supplementary Fig. 6** and l. 192-195).

Despite the obvious advantages of using an organism that is easy to manipulate and for which many genomic and genetic resources are available, we were also motivated by the observation that *Chlamydomonas* is one of the most abundant algal genera found in the site from which the earlier bacterial culture collection from *A. thaliana* was established (Cologne Agricultural Soil, CAS). In the revised version of the manuscript, we now include the results of a survey of natural soil samples using the eukaryotic *18S* rRNA gene that illustrates this fact (please, see new **Supplementary Fig. 3** and our response to point 2.5 below).

1.3. More importantly, *Cr* is not related to the charophyte algae from which land plants evolved (see for example Delwiche & Cooper 2015 *Curr Biol* 25(19):R899-910. doi: 10.1016/j.cub.2015.08.029) and so there is little reason to suppose that there are evolutionarily conserved mechanisms in terms of interactions with bacteria, other than between eukaryotes generally. On lines 105-108, the authors describe the selection of bacterial taxa representative of 'phylogenetically diverse plant species' - and yet these are compared with just one algal species that is likely far more distantly related to charophytes than lycopods are to angiosperms. Throughout the text there are statements that imply that *Cr* CC1690 is representative of green algae (eg lines 25-26, 117-118, 301, 317-318, 333-334, among others), providing a misleading impression. Indeed the text on lines 117-118 extrapolates *Cr* as representing a wide range of photosynthetic unicellular eukaryotes, belying the diversity across the eukaryotic tree of life.

We would like to thank the reviewer for raising this important issue.

To test whether the process of phycosphere microbiota assembly is also found in other taxonomic lineages of green algae besides the chlorophyte *Cr*, we performed a new experiment where we characterized the soil-derived phycosphere microbiota of a selection of subaerial algal strains, isolated from the same terrestrial environment. This group of algae includes chlorophytes, from the classes Trebouxiophyceae (*Microthamnion*), and Chlorophyceae (*Chlamydomonas*), as well as streptophytes belonging to the Klebsormidiophyceae (*Klebsormidium*), and Zygnematophyceae (*Spiroglea*) classes, the latter of which has the most recent common ancestor with embryophytes of all green algae (Cheng *et al.*, 2019). Remarkably, the results from this survey show that different microalgae can assemble distinct soil-derived microbial communities in a pattern of diversification that relates to the host phylogeny (new **Supplementary Fig. 6**). Furthermore, the phycosphere communities of all characterized algal strains were dominated by members of the core bacterial taxa shared with the root microbiota of land plants. The results from this experiment indicate that assembly of distinct phycosphere communities from soil is not specific of *Cr* but can be also observed for other subaerial green algae from diverse taxonomic groups. Future work will be aimed at elucidating the ecological and molecular mechanisms that cause this diversity in soil-borne phycosphere communities and their physiological relevance for their hosts. We have added the results from this experiment to the revised version of the manuscript in l. 181-199 and **Supplementary Fig. 6**.

In addition, we agree with the reviewer that in the previous version of the manuscript we made generalizations extrapolating *C. reinhardtii* as a representative of a wide range of photosynthetic unicellular eukaryotes. Although the results from the experiment outlined above add generality to some of our claims, we have nonetheless corrected this throughout the text to avoid making this unjustified generalization.

1.4. I was not clear about the purpose of whole genome sequencing work, since it shows similar results to the 16S metabarcoding. As mentioned above, the correlations between the composition of the microbiota may simply be just that. The WGS data might have provided the means to assess metabolic capabilities between those taxa that were found associated compared to the soil bacteria generally but there was no attempt to do that. Incidentally the ENA accession number does not provide access to the sequence data.

Our first motivation for sequencing the genomes of our core culture collection of *Cr*-associated bacteria was to test if there were significant differences in terms of functions encoded in their genomes compared to those from the collections derived from *A. thaliana* and *L. japonicus* roots. Surprisingly, our analyses yielded predominantly negative results, and no significant differences in terms of KEGG functional categories were identified. A subset of these negative results is presented in the main text (l. 226-230). Finally, we predict that identification of different metabolic capabilities between the phycosphere and soil microbiomes could be more appropriately explored in future studies using shotgun metagenomics, given that we do not currently have a comparable culture collection of soil bacteria from the same site. The sequencing data can now be publicly accessed through the ENA repository (PRJEB43117).

1.5. Lines 307-310 - it is not just cell wall components that might be present in the phycosphere. *Cr* secretes a range of fermentation products at night including formate, acetate and H₂.

We agree with the reviewer and have added this information to the corresponding sentence in the discussion (l. 361-365).

1.6. lines 141-143 - it is not clear what this experiment is showing - which fraction is being measured?

This experiment shows the assembly of distinct bacterial communities by *C. reinhardtii* in a mesocosm system. Here, we compare bacterial community profiles of samples obtained from liquid cultures in flasks containing organic carbon-free media that were co-inoculated with soil bacterial washes and axenic *Cr* cultures. We have now rephrased the corresponding text to improve clarity (l. 166-170).

Reviewer #2 (Remarks to the Author):

2.1. The paper is well written and undoubtedly shows that *Chlamydomonas* can alter a bacteria community whether in soil or in a syncom. However, I don't really know what this means as adding a photosynthetic eukaryote to any bacterial community will result in changes to the community.

We would like to thank the reviewer for their comments on our manuscript.

We agree with the reviewer that adding a photosynthetic organism to a soil bacterial community will likely result in changes to said community. Characterizing these changes, and comparing them with those induced by land plants, is one of the main motivations of our study. Additionally, we have performed a new experiment including a taxonomically diverse set of terrestrial microalgae (see also response to point 1.3), which shows that phycosphere community structure is host-species specific, and varies in a pattern that mirrors the phylogeny of the host.

Furthermore, to directly test whether phycosphere community establishment could also be observed without adding a photosynthetic eukaryote to the soil community, we performed an experiment using natural soil, where native microalgae proliferated with minimal intervention (see also our response to point 2.5 below). The results from this experiment show changes in soil community composition, which are concurrent with the proliferation of photosynthetically active chlorophyte algae on the soil surface, and are similar to those observed in our *C. reinhardtii* inoculation experiments (new **Supplementary Fig. 3**).

2.2. I also accept that while diffusion of carbon and probably other metabolites from the algae is important there are proximity effects i.e. when the algae are present with they community they shift is structure (albeit very slightly Figure 5c). However, diffusion gradients and macromolecular shedding might cause similar effects.

We thank the reviewer for pointing this out. We have now modified the discussion to mention the possibility that diffusion gradients and / or macromolecular shedding, besides diffusion of organic carbon or other metabolites, might cause the observed changes in community composition and explain some of the patterns that we observed in our split-system experiment (l. 403-405).

2.3. I also accept that the communities assembled by the algae and plants have similarities although also big differences. However, again that is hardly surprising as the same soil community is the ultimate source of the bacteria.

To test this, we have performed an analysis where we simulated the process of recruitment of bacterial species from the starting soil microbiota by *Cr* using the hypergeometric distribution. This analysis shows that the observed overlap between the *Cr* phycosphere and *At* root communities (**Fig. 1c-d**) is highly significant ($P=3.06 \times 10^{-33}$) and cannot be explained by a random process of microbial recruitment from the same starting soil community. We now describe the outcome of this analysis in the results section (l. 74-78).

In addition, we would like to point out that one of our experiments was performed using bacteria derived from two distinct soils (Cologne and Golm soils). The results from this experiment show that the phycosphere communities assembled by *C. reinhardtii* from either soil were highly similar (green points of different shapes in **Fig. 2a**), and in both cases a significant overlap with the abundant taxa found in the root microbiota was identified.

2.4. A much more difficult question is whether the algae really alters the bacterial community in a meaningful way in a natural setting and whether there is any mechanistic interaction between the algae and the bacterial community. This is clear with plant roots where it can be shown that bacteria attach to roots and form biofilms.

We agree with the reviewer that exploring how green algae alter soil bacterial communities in natural terrestrial ecosystems, as well as studying the mechanisms underlying their interactions are fundamental questions that remain open and should be addressed in future research. We now mention this explicitly in the discussion (l. 412-415).

In addition, we would like to point out that biofilm formation and attachment to the root is not necessary for the establishment of the rhizosphere microbiota, which has been shown to be also driven by release of soluble organic carbon compounds and other metabolites to the surrounding soil (e.g., Bulgarelli *et al.*, 2013; Philippot *et al.*, 2013). It is therefore reasonable to hypothesise that a similar process might take place for soil-borne microalgae and their associated phycosphere communities.

2.5. Flooding soil with *Chlamydomonas* and measuring changes does not look like a natural situation to me.

To address this point we have conducted a new experiment with natural soil, where we did not perform a food-inoculation using *Chlamydomonas*. Instead, we analysed changes in soil microbial community composition after exposing the natural soil to a light/dark cycle, which induced the proliferation of native microalgae without additional intervention. After incubation for 7 weeks, we obtained samples from the soil surface, measured chlorophyll content, and performed eukaryotic *18S* and bacterial *16S* rRNA profiling. As negative controls, we also obtained samples from the initial soil, as well as from soil which was kept in the dark but otherwise under the same conditions throughout the duration of the experiment. We observed a significant increase in chlorophyll fluorescence in the soil surface exposed to a light/dark cycle compared to the negative controls, which was concomitant with a significant increase in the relative abundance of chlorophyte algae on the soil surface when exposed to light (new **Supplementary Fig. 3a-b**).

Interestingly, we observed that the genus *Chlamydomonas* was one of the most abundant eukaryotic taxa in the light-exposed soil, including several ASVs assigned to *C. reinhardtii*. Together with this significant proliferation of green algae and increase in chlorophyll content, we observed a clear differentiation of the bacterial communities from the lighted soil surface compared to those of the initial soil and negative controls, which clustered together (**Supplementary Fig. 3d**). Importantly, the bacterial communities associated with these naturally occurring algae were dominated by members of the core root and phycosphere taxonomic groups, which were significantly less abundant in the initial soil and negative controls (**Supplementary Fig. 3e**). Additionally, comparison with *16S* profiles obtained from the greenhouse experiment (**Fig. 1**) revealed that the communities found on the light-

exposed soil surface resembled those from the *Cr* phycosphere and, together with samples of the rhizosphere and root of *At*, separated from the initial soil and dark soil surface samples along the second principal component (**Supplementary Fig. 3c**). Together, these new data show that phycosphere communities assembled using flood inoculation of axenic *Chlamydomonas* cultures resemble patterns of microbial diversity observed in associations with naturally occurring green algae in a natural soil.

minor points

2.6. -The use of symbol sizes in figure 2 and support fig 4 is pretty much impossible to distinguish in the figure

We have now increased the differences between the symbol sizes.

2.7 -The legend in figure three appears to be scrambled, panel a is really c, b is d, c is a and d is b.

We have now corrected this mistake.

Reviewer #3 (Remarks to the Author):

In this manuscript, Durán, Flores-Uribe et al. describe the microbiota of the algae *Chlamydomonas* and compare its taxonomic composition and genomic features to the plant root microbiota. The authors use publicly available data on the microbiota of a variety of plants, in addition their own data from previous studies on the model plant *Arabidopsis*, and identify core taxa that are present in all hosts. Moreover, the authors establish an extensive culture collection of *Chlamydomonas*-associated bacteria and perform synthetic community (SynCom) experiments. In this study, the authors performed experiments with single SynComs and competition between SynComs of *Chlamydomonas* and *Arabidopsis*. Depending on the experiments, the SynCom consisted of up to 26 strains. The SynComs are designed to contain equivalent taxa (9 strains from each host), and were able to assemble communities with similar composition on both hosts. During competition of both SynComs, strains performed better in their “native” host compared to strains isolated from the other host. To explore whether the release of diffusible photoassimilates from the algae could explain community composition, the authors performed split co-culture experiments and analyze algal growth and community composition. The communities were different when *Chlamydomonas* and bacteria could only exchange metabolites compared to when both were in physical contact to each other.

Overall, the manuscript is well written and the figures are beautifully crafted. The study is original, and of particular interest to the plant and algae research communities. *Chlamydomonas* is a model organism and its research in this context fosters our understanding of host-associated microbiota assembly. Foremost, the new culture collection provides a valuable tool for future studies. The experimental procedures are sound and the authors have a long-standing experience in 16S rRNA amplicon data analysis. The experiments are well designed and give important insights in microbiota assembly.

We would like to thank the reviewer for their positive evaluation of our work and their constructive comments.

Still, I have some comments that would improve the manuscript:

3.1. I wonder what defines “the phycosphere community” when the authors summarize the results of the split co-culture experiment? The authors show differences in community compositions but no details how the communities differ. Surely, the communities are different and physical contact has an additional influence, but how does this difference compare to the difference of phycosphere community composition between soil, liquid shake flask and with *At* strains? I would suggest that the authors take this into account when they seemingly talk about “the” phycosphere community (e.g. line 23) and maybe rephrase.

We thank the reviewer for raising this point. In the previous version of the manuscript we did not explain in detail the differences between the phycosphere communities from the split co-culture experiment or related these differences to the differences of phycosphere community composition found in the soil and liquid flask systems. We observed that differences between the SC+C|– and SC|C communities in the split system (**Fig. 5a**) were mainly explained by a significant enrichment of Sphingomonadaceae and Comamonadaceae when the SynCom was growing in the same compartment as *Cr* (SC + C), and a depletion of Pseudomonadaceae. Despite differences among experimental systems, these patterns mirror the results from the natural soil, mesocosm, and SynCom flask experiments (**Figs. 1-3**), where Sphingomonadaceae was one of the most abundant taxa in the *Cr* phycosphere and Pseudomonadaceae dominated the bacterial communities in the absence of *Cr*. We have now added this additional information to the corresponding section of the results (l. 296-303).

3.2. Similarly, I stumbled over the “core” ecological principles (line 24) as being a very vague and general expression, especially since no further explanation is given what it actually means in the specific context of the study. Do the authors mean that the “core” (or basic) principle is provision of nutrients/metabolites by the host in general? Or that similar nutrients are provided by *Arabidopsis* and *Chlamydomonas*? Or that nutrients and contact are required for a fully differentiated microbiota assembly in *Arabidopsis* and *Chlamydomonas*? Please clarify.

We agree with the reviewer and have corrected this in the revised version of our manuscript. Instead, we discuss the different processes that might explain the observed overlap between the root and phycosphere communities, such as provision of metabolites, physical contact, or exchange of molecular signals (l. 392-415).

3.3. I have a question regarding Supp. Fig. 2b-d and Fig.1c. Theoretically, the same ASV could be enriched in the root and phycosphere. Looking at Supp. Fig. 2b-d, this does not seem to be the case (or

only for very low abundant strains). However, there seem to be some commonality between root and phycosphere (Fig1c). Is this similarity only seen at OTU level as presented in Fig.1c? Maybe the authors could explain in the text why they switched the taxonomic level for different analysis.

The reviewer is correct in that root- or phycosphere-enriched ASVs are mostly compartment-specific. This could explain the separation between root and phycosphere samples shown in **Fig. 1b**. Nonetheless, the similarities between these two compartments, illustrated in **Fig. 1c**, can be explained by the presence of shared ASVs, which are not specifically enriched in either compartment, and that account for approximately 50% of the communities.

We should also clarify that the analyses shown in **Supplementary Fig. 2b-d** and **Fig. 1b-c** are both based on ASVs, while only the analysis shown in **Fig. 1d** is based on OTUs. The reason for switching to this higher taxonomic level is to be able to directly compare with earlier data from Bai *et al.*, 2015, where the *A. thaliana* culture collection was described. We now explain the rationale for using these two types of analysis (l. 882-884) and mention in each figure caption at what level was the analysis was performed.

3.4. In Fig.1d, I was surprised to see Firmicutes clustering with the Alphaproteobacteria in the phylogenetic tree based on 16S sequences. It seems to be different when using AMPHORA genes. I guess it is correct, but it was just unexpected to me.

The phylogeny shown in **Fig. 1d** is based on a sequence alignment of the V5-V7 hypervariable region of *16S* rRNA gene obtained from culture-independent profiling of natural communities. The placement of the Firmicutes with the Alphaproteobacteria is likely caused by the fact that this fragment of the *16S* gene is relatively short and contains only limited information for phylogenetic analyses, particularly across large evolutionary distances. Contrary to this, the tree depicted in **Supplementary Fig. 7b** is based on a concatenated alignment of the 32 AMPHORA marker genes extracted from the whole-genome assemblies of our cultured isolates. The tree based on this sequence alignment, which contains more information, correctly separates the Firmicutes from the Proteobacteria. We now clarify in the caption of **Fig. 1d** that this phylogeny was inferred from *partial* 16S rRNA sequences.

3.5. Comment: Regarding the experiments of Supp. Fig4b and Fig.5, a potential future experiment could be to mimic physical contact by heat killed *Chlamydomonas* and adding the Artificial Photosynthates or using the split co-culture.

We thank the reviewer for this suggestion. We are currently planning similar experiments using the split co-culture systems to explore how physical contact influences phycosphere community composition and functions.

3.6.) I appreciate the time-course experiments to monitor community assembly. In Fig.3d, why did the authors include the rel. abundance of *Chlamydomonas* in the graph? While it is interesting to see that

Chlamydomonas abundance is as high as entire microbiota, the community composition of Input, SynCom alone and Cr+SynCom cannot be compared anymore.

We agree with the reviewer and have removed the *Chlamydomonas* relative abundance from the plot and re-calculated the relative abundances of the bacterial strains, now shown in **Fig. 3d**.

3.7. Fig. 4cd show the aggregated relative abundance. As this is the average of all strains, I wonder whether this is mainly driven by few strains or if most strains show such a host-specificity?

Our analyses show that the majority of the SynCom members show signatures of host-preference either in the FlowPot or in the liquid flask system, and the strongest effects were found for members of the Comamonadaceae and Xanthomonadaceae families. Due to space limitations, we do not discuss these results in more detail in the main text.

3.8. In multiple experiments, the authors use *Chlamydomonas* fluorescence to assess its abundance (e.g. Fig5d, Fig3b). As bacteria can also release fluorescent compounds (e.g. siderophores), I wonder whether it is a suitable method to monitor changes in Cr abundance when comparing presence and absence of a SynCom. Measuring the fluorescence of a bacterial supernatant filtrate (+/- *Chlamydomonas*) would reveal if there is a background due to secretion of bacterial compounds.

We have performed these tests while developing the different experimental systems and observed very low bacterial background when measuring fluorescence by using 440 nm / 680 nm Ex / Em wavelengths, which we employ to detect the presence of chlorophyll in a sample. As an example, we show below the fluorescence measurements of the bacteria-only compartments of the split co-culture system, which we performed as negative controls but do not show in main **Fig. 5d**.

Rebuttal Fig. 1. Relative chlorophyll fluorescence measurements from the split system experiment. This figure extends main **Fig. 5d** including measurements from the negative controls, i.e., compartments containing bacteria and axenic medium.

3.9. Throughout the manuscript, the figure legends mostly interpret the results and only provide minimal technical information. Personally, I rather prefer if the legends states the experimental settings, sample number, statistics etc. to make it possible to understand how the data shown was generated. Often, the method section does not cover every experimental detail or small variations of the specific experiment shown in the respective figure.

We thank the reviewer for this suggestion and have now added sample numbers and information regarding the statistics in each figure caption. Furthermore, we have now compiled a list of the main experiments and a summary of their technical details and sample numbers (new **Supplementary Table 1**, see also **Supplementary Fig. 1**). In the revised version of the manuscript, we relate each figure to their corresponding experiment ID in this supplementary table and provide the corresponding references in the Methods section.

3.10. In the discussion of the data presented in Figure 4 and 5, the authors follow the model of sequential selection of microbiota members by the host that gets increasingly specific. Considering the “reciprocal complementation” of both SynComs on family level, the host (or ecological factors) applies a rough filter to select on a “high taxonomic level”. Only in direct competition between close relatives, strain-specific adaptations to their respective host become observable. I totally agree with this argumentation, but I got a bit confused because first the authors highlight differences between extracellular organic carbon released from *Arabidopsis* and *Chlamydomonas* (307 f) and later suggest the release of photoassimilates as first principle driving community assembly in these hosts (339 f). I would assume a comparison of exudates from both organism could be insightful to clarify these questions.

We have modified the relevant section of the discussion to clarify our interpretation of the data (l. 392-415). We hypothesise that the release of organic carbon compounds by the host constitutes the first step in selection of a subset of heterotrophic bacteria from the soil. Differences in exudate composition as well as other physiological differences, such as the presence of an advanced immune system in *A. thaliana*, could then contribute with host species-specific signatures in community composition at a higher taxonomic resolution.

We also agree with the reviewer that a comparative analysis of exudates could also be insightful. We are currently planning such experiments and developing protocols for metabolomic profiling of exudates from both *A. thaliana* as well as *C. reinhardtii* when grown axenically or co-inoculated with different bacterial SynComs. However, we believe these experiments will require substantial time and effort and consider them beyond the scope of the current study.

Minor points:

3.11. The At and Cr SynComs “assemble into taxonomically equivalent communities”. If the input is taxonomically equivalent, all strains will be detected in the end. Maybe mention that the relative abundance/composition rather taxonomy is similar.

We have now corrected this in the text and use the term ‘taxonomically similar’ instead, as suggested by the reviewer (l. 20-22 and 244).

3.12. Fig.3: the description in the legend does not correspond to the correct panels.

We have now corrected this mistake.

3.13. As some beta-diversity analysis are done on ASV, OTU and family-level, it would be important to mention in each figure legend what level was analysed (e.g. Fig5a).

We now mention in each figure caption at what level the analyses were performed.

REVIEWERS' COMMENTS

Reviewer #1 (Remarks to the Author):

The authors have done an excellent job in addressing the comments of the reviewers. Particularly noteworthy is the fact that they took on board the need to relate their studies to the natural environment and carried out extensive further experiments with soil microbial communities. There are a few minor issues that would be worth correcting:

Line 48 - spell out *Chlamydomonas* in Abstract and again at first mention in text

Line 97 - delete 'them'

Line 102 - change 'on' to 'of'

Line 156 and elsewhere - 16S does not need to be italicised

Line 185 - delete the comma

Line 198 - should read 'not specific for Cr'

Line 240 - delete the comma

Line 255 - should be 'allowed them to colonize'

Line 343 - perhaps 'are consumed by heterotrophic bacteria'?

Lines 361-365 - this sentence is long and hard to follow - suggest splitting it

Line 392 - change 'Carbon' to 'Fixed carbon'

Reviewer #3 (Remarks to the Author):

The authors have satisfactorily addressed all issues raised during the revision. Great to see the additional experiment showing that algae from natural soil assemble similar microbiota as the model laboratory strain. I also appreciate the table giving an overview of all experiments performed in this study.

Minor issue:

Figure 5 uses the symbol Cr $[-]$ for Cr only control, but in the main text Cr $[-]$ is used. Please change to make it consistent.

We would like thank the referees for their constructive comments. We provide below a point-by-point response to their remaining minor issues and suggestions. Please, also note that we have moved Supplementary Fig. 4 to the main text (now main Fig. 2), as we consider this figure complements the results from our natural community experiments (shown in Fig. 1).

Reviewer #1 (Remarks to the Author):

The authors have done an excellent job in addressing the comments of the reviewers. Particularly noteworthy is the fact that they took on board the need to relate their studies to the natural environment and carried out extensive further experiments with soil microbial communities.

We thank the reviewer for their positive evaluation of the revised version of the manuscript and their constructive suggestions.

There are a few minor issues that would be worth correcting:

Line 48 - spell out Chlamydomonas in Abstract and again at first mention in text

We have implemented this change.

Line 97 - delete 'them'

We have corrected this.

Line 102 - change 'on' to 'of'

We have corrected this.

Line 156 and elsewhere - 16S does not need to be italicised

We have removed the italics throughout the text.

Line 185 - delete the comma

We have implemented this change.

Line 198 - should read 'not specific for Cr'

We have corrected this.

Line 240 - delete the comma

We have implemented this change.

Line 255 - should be 'allowed them to colonize'

We have corrected this.

Line 343 - perhaps 'are consumed by heterotrophic bacteria'?

We have implemented this change.

Lines 361-365 - this sentence is long and hard to follow - suggest splitting it

We have simplified this sentence.

Line 392 - change 'Carbon' to 'Fixed carbon'

We have changed 'Carbon' to 'Organic carbon'.

Reviewer #3 (Remarks to the Author):

The authors have satisfactorily addressed all issues raised during the revision. Great to see the additional experiment showing that algae from natural soil assemble similar microbiota as the model laboratory strain. I also appreciate the table giving an overview of all experiments performed in this study.

We thank the reviewer for their positive comments on the revised version of the manuscript.

Minor issue:

Figure 5 uses the symbol Cr [] for Cr only control, but in the main text Cr - is used. Please change to make it consistent.

We have fixed this issue.